# Improved ANAP incorporation and VCF analysis reveal details of P2X7 current facilitation and a limited conformational interplay between ATP binding and the intracellular ballast domain

**Anna Durner[1], Ellis Durner[2], Annette Nicke[1]\***

[1]Walther Straub Institute of Pharmacology and Toxicology, Faculty of Medicine, LMU Munich, Munich, Germany; [2]Lehrstuhl für Angewandte Physik und Center for Nanoscience, LMU Munich, Munich, Germany

**Abstract** The large intracellular C-terminus of the pro-inflammatory P2X7 ion channel receptor (P2X7R) is associated with diverse P2X7R-specific functions. Cryo-EM structures of the closed and ATP-bound open full-length P2X7R recently identified a membrane-associated anchoring domain, an open-state stabilizing "cap" domain, and a globular "ballast domain" containing GTP/GDP and dinuclear $Zn^{2+}$-binding sites with unknown functions. To investigate protein dynamics during channel activation, we improved incorporation of the environment-sensitive fluorescent unnatural amino acid L-3-(6-acetylnaphthalen-2-ylamino)–2-aminopropanoic acid (ANAP) into *Xenopus laevis* oocyte-expressed P2X7Rs and performed voltage clamp fluorometry. While we confirmed predicted conformational changes within the extracellular and the transmembrane domains, only 3 out of 41 mutants containing ANAP in the C-terminal domain resulted in ATP-induced fluorescence changes. We conclude that the ballast domain functions rather independently from the extracellular ATP binding domain and might require activation by additional ligands and/or protein interactions. Novel tools to study these are presented.

**\*For correspondence:**
annette.nicke@lrz.uni-muenchen.de

## Editor's evaluation

This manuscript constitutes a valuable foray into the conformational rearrangements throughout various domains of the notoriously difficult-to-study P2X7 receptor, with a focus on the enigmatic intracellular 'ballast' domain. The molecular origin of the facilitation process and effects by intracellular factors will require future study, but the authors provide convincing evidence that the ballast domain is unlikely to undergo major conformational changes upon ATP-induced gating. The work is of interest to those interested in the role of enzymatically active intracellular domains of membrane proteins.

## Introduction

P2X receptors (P2XR) are trimeric non-selective cation channels that are activated by extracellular adenosine triphosphate (ATP). The structure of a single P2X subunit has been compared to a dolphin, with two transmembrane domains (TM1 and TM2) that form the 'fluke', and a large extracellular domain, comprising the 'body', left and right 'flippers', and a 'head' domain that reaches over one of three inter-subunit ATP binding sites (*Kawate et al., 2009*). The intracellular N- and C-termini are

short in most P2X subtypes and have only been resolved in the open state of the P2X3R and, more recently, in the open and closed states of the P2X7R (*Mansoor et al., 2016*; *McCarthy et al., 2019*). The pro-inflammatory P2X7 subtype is expressed in immune cells and considered an important drug target. In contrast to the other P2XR family members, it has a low ATP sensitivity, shows complete lack of desensitization, and contains a large intracellular C-terminus (240 amino acids [aa]), which mediates diverse downstream effects such as interleukin secretion, plasma membrane permeabilization, blebbing, phosphatidylserine flip, and cell death (*Kopp et al., 2019*). The recently determined cryo-EM structures of the full-length rat P2X7R in the apo/closed and ATP-bound open states (*McCarthy et al., 2019*) did not only elucidate details of P2X desensitization, but finally unveiled the structure of the large P2X7 C-terminus. Accordingly, intertwined β-strands from all three subunits form an open state-stabilizing 'cap domain', that was also found in the P2X3R (*Mansoor et al., 2016*). In the P2X7R, however, this 'cap' is stabilized by a highly palmitoylated membrane-associated 'Cys-anchor' domain, which prevents desensitization. The remaining residues 393–595 fold into a dense globular structure (the so-called 'ballast domain'), which contains a novel guanosine nucleotide binding motif and a dinuclear zinc binding site. A stretch of 27–29 aa (S443-R471) was not resolved, and it is unclear if each ballast domain is formed by a single subunit or if a domain swap occurs between subunits (*McCarthy et al., 2019*). While these structures represent a milestone in P2X7 research, the transition dynamics between receptor states in a cellular environment as well as the molecular function of the ballast domain and how it is affected by ATP binding remain unclear. Likewise, the molecular mechanism of current facilitation, a P2X7-characteristic process that describes faster and/or increased current responses upon repeated ATP application, is not understood. In this study, we set out to determine conformational changes associated with P2X7-specific functions by voltage clamp fluorometry (VCF). This method allows simultaneous recording of current responses and associated molecular movements that are reported by an environment-sensitive fluorophore. We have previously used site-specific cysteine-substitution and the thiol-reactive fluorophore tetramethyl-rhodamine-maleimide (TMRM) to show a closing movement of the head domain during activation of the oocyte-expressed P2X1R (*Lörinczi et al., 2012*). However, this procedure is limited to extracellularly accessible residues. To investigate intracellular rearrangements, we therefore employed the fluorescent unnatural amino acid (fUAA) L-3-(6-acetylnaphthalen-2-ylamino)–2-aminopropanoic acid (ANAP) (*Lee et al., 2009*). This can be site-specifically incorporated into a protein by repurposing the *amber* stop codon (TAG) and introducing a corresponding suppressor tRNA (CUA anticodon) loaded with ANAP. A plasmid encoding an ANAP-specific bio-orthogonal suppressor tRNA/aminoacyl-tRNA synthetase pair (*Chatterjee et al., 2013*) has been obtained by co-evolution and selection (*Lee et al., 2009*) and was successfully used to study voltage-gated and ligand-gated ion channels (*Andriani and Kubo, 2021*; *Kalstrup and Blunck, 2018*; *Kalstrup and Blunck, 2013*; *Soh et al., 2017*; *Wulf and Pless, 2018*). This stop-codon suppression can, however, lead to premature translational termination or aberrant stop-codon substitution (read-through) (*Braun et al., 2020*; *Kalstrup and Blunck, 2017*; *Klippenstein et al., 2018*; *Pless et al., 2015*; *Poulsen et al., 2019*).

Here, we provide an improved method for fUAA incorporation into *Xenopus laevis* oocyte-expressed proteins and analyzed membrane surface expression and functionality for a total of 61 P2X7R mutants with ANAP substitutions in the extracellular head domain, the second transmembrane domain (TM2), and the intracellular N- and C-termini. Using VCF, we identified 19 positions in which ANAP reported ATP-induced localized rearrangements. To further expand the VCF toolbox, we demonstrate simultaneous recordings of fluorescence changes from ANAP in combination with other fluorophores. We conclude from our data that (i) current facilitation is intrinsic to the P2X7 protein and likely caused by a change in gating and (ii) the cytoplasmic ballast functions rather independently from the extracellular ligand binding domain and might require activation by additional ligands or protein interactions.

## Results
### Improved ANAP incorporation by cytosolic co-injection of mutated *X. laevis* eRF1 cRNA

To implement and optimize a protocol for incorporation of ANAP into *Xenopus* oocyte-expressed protein, we initially used the P2X1R as it was already intensively studied in our lab (*Lörinczi et al., 2012*) and has functional similarity with the P2X3R, which at the beginning of this study, represented

the only P2XR for which the intracellular termini were resolved (*Mansoor et al., 2016*). Using the original 2-step-injection protocol (*Kalstrup and Blunck, 2017*; *Kalstrup and Blunck, 2013*) and a simplified procedure where all components required for the expression of UAA-containing receptors are injected simultaneously (*Figure 1A and C*), we introduced ANAP into non-conserved positions within the N-terminally His-tagged P2X1R N- and C-termini (position 10 and 388, respectively, ANAP substitutions indicated by *) and compared the formation of full-length and truncated receptors in the plasma membrane by SDS-PAGE. As seen in *Figure 1B*, ANAP-containing P2X1Rs were efficiently expressed and virtually no read-through product was detected in the absence of ANAP. The new protocol resulted in less variable protein expression but also a reduced ratio of full-length and truncated His-P2X1 EGFP protein (*Figure 1D*). The relative amount of full-length protein was neither increased by different forms of ANAP application nor by variation of injection protocols (*Figure 1—figure supplement 1A and B*). Therefore, we tested if a mutated eukaryotic release factor (eRF1(E55D)), which was previously shown to favor UAA-incorporation over translational termination in HEK293T cells (*Gordon et al., 2018*; *Schmied et al., 2014*) could also be used in the *Xenopus* oocyte expression system. Indeed, co-injection of either purified *X. laevis* eRF1(E55D) protein (*Figure 1—figure supplement 1B*) or the respective *in vitro* synthesized cRNA (*Figure 1C and D*) resulted in a more than threefold higher ratio of full-length and truncated receptor constructs compared to the 1-step injection method without eRF1(E55D) and a smaller standard deviation compared to the 2-step injection method (1-step+eRF1(E55D): 1.469±0.229; 1-step: 0.418±0.082; 2-step: 1.603±0.933; mean ± S.D.). The applicability of this approach was confirmed for the hα1 glycine receptor (GlyR) A52* mutant (*Figure 1—figure supplement 1C*; *Soh et al., 2017*). In conclusion, this optimized protocol led to more reproducible expression and increased formation of full-length ANAP-labeled receptors and was used in all following experiments.

## Evaluation of plasma membrane expression of full-length ANAP-containing P2X7Rs

Next, we incorporated ANAP into the P2X7R in sites chosen based on previous structure-function studies and the cryo-EM structures (*McCarthy et al., 2019*). As a positive control, we first introduced ANAP into the head domain (*Figure 2A and B*), which is known to undergo substantial movements and/or ligand interactions with clear changes of TMRM fluorescence in the P2X1R (*Lörinczi et al., 2012*) and P2X7R (*Figure 2—figure supplement 1A*). Next, based on the comparison of the P2X4 and P2X3 crystal structures in the open and closed states (*Hattori and Gouaux, 2012*; *Kawate et al., 2009*; *Mansoor et al., 2016*), and the identification of the human P2X7 channel gate and selectivity filter around residue S342 (*Pippel et al., 2017*), we selected positions in the second transmembrane helix. Finally, we introduced ANAP throughout the intracellular region in positions that we suspected to undergo conformational changes upon channel activation, as well as in six positions in the unresolved 29 aa stretch. As shown in *Figure 2C*, all constructs with ANAP substitutions in the N-terminus and the head domain as well as three out of four constructs with substitutions in TM2 were formed in full length, indicating that receptors that are truncated before or within TM2 are retained in the endoplasmic reticulum and likely undergo degradation. Interestingly, ANAP incorporation into G338 completely prevented membrane incorporation while cysteine substitution in the equivalent position of human P2X7R led previously to surface-expressed, but non-functional receptors (*Pippel et al., 2017*).

Starting from T357 in the C-terminus, introduction of the *amber* stop codon resulted in variable ratios of truncated and full-length receptors. Surface expression of full-length receptors was particularly low for constructs containing ANAP in the C-terminal cap (K387*, C388*) and ballast (I577*) domains, while it was most efficient for ANAP-substitutions in positions 517–537 (in particular L527* and E537*) and in the very C-terminus (Y595* and 596*).

In summary, most substitutions within the C-terminus led to a dominant formation of truncated P2X7 protein besides full-length receptors. Nevertheless, the majority of these constructs showed clear current responses (*Figure 2C*, *Table 1*). Since the truncated forms were not expected to interfere with the fluorescence signal, functional constructs that were expressed at least partly in full length were further analyzed by VCF.

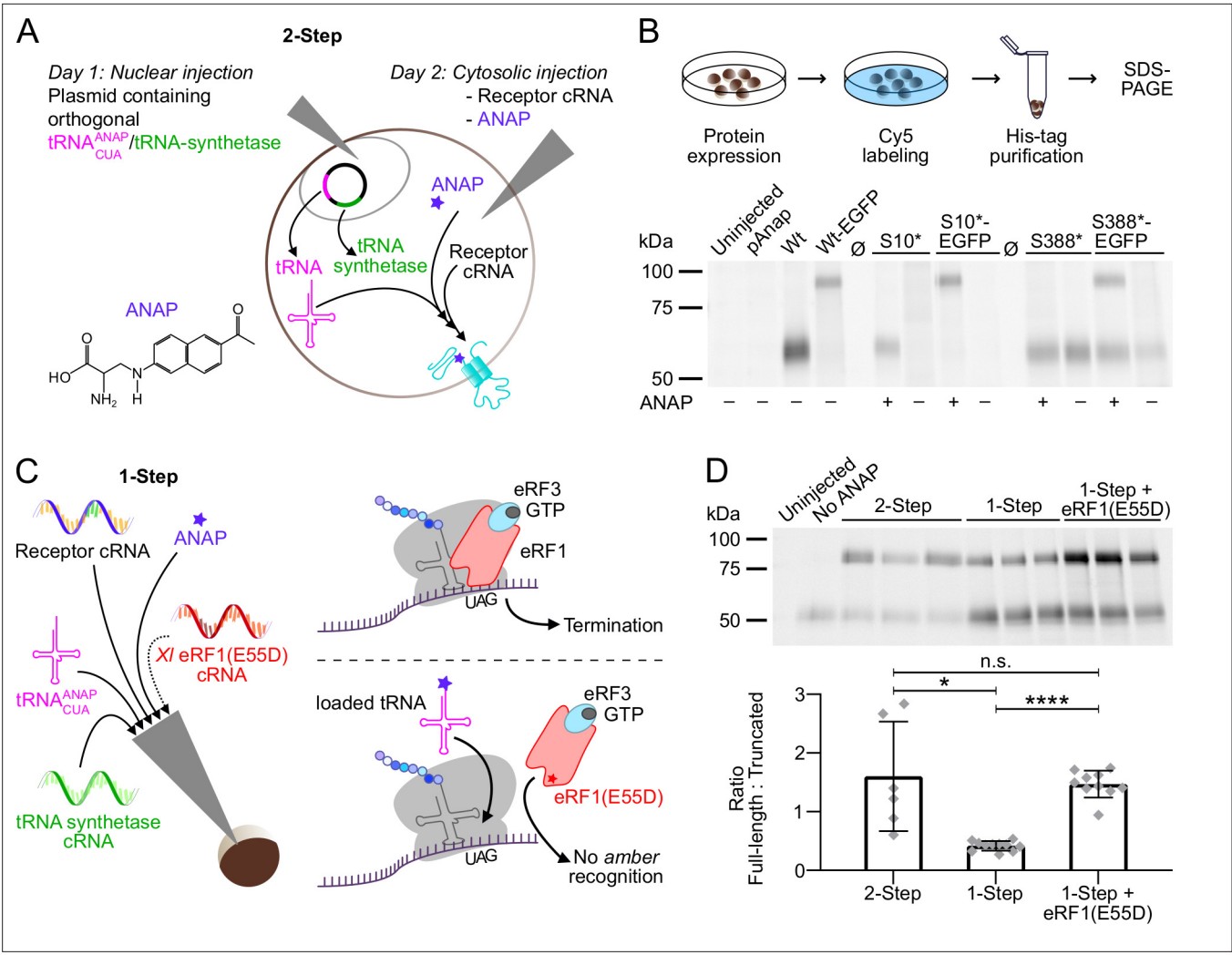

**Figure 1.** Optimization of fluorescent unnatural amino acid (fUAA) incorporation into *Xenopus laevis* oocyte-expressed P2X1 receptor. (**A**) Molecular structure of L-3-(6-acetylnaphthalen-2-ylamino)–2-aminopropanoic acid (ANAP) and schematic representation of the 2-step injection method for site-specific ANAP incorporation using the *amber* stop-codon (UAG) and a plasmid containing the orthogonal tRNA/tRNA-synthetase pair. (**B**) Representative SDS-PAGE analysis of plasma membrane-expressed ANAP-labeled (S10* or S388*) rat P2X1Rs (46 kDa without glycosylation). A C-terminal EGFP-tag (27 kDa) was added as indicated to enable detection of premature termination at position 388. Oocytes were injected as shown in A and labeled with membrane impermeable Cy5-NHS ester. His-tagged P2X1Rs were extracted in 0.5% *n*-dodecyl-β-D-maltoside, purified via Ni²⁺-NTA agarose, and separated by SDS-PAGE (8%). Noninjected oocytes and oocytes injected only with the plasmid pANAP, P2X1 cRNA without the *amber* stop codon (Wt), or without ANAP (as indicated) served as controls. Note, that twice the amount of protein was loaded for P2X1(S10*). Ø indicates empty lanes. Two to three independent experiments were performed. (**C**) Representation of the 1-step injection method and all components required for UAA-labeling plus optional *X. laevis* eRF1(E55D) cRNA (left) and (right) scheme of protein translation termination by eRF1 (upper panel) and how overexpression of the mutated form of eRF1 favors *amber*-encoded fUAA incorporation by outcompeting endogenous eRF1 (lower panel). (**D**) Comparison of Cy5-labeled membrane-expressed full-length and truncated His-rP2X1-EGFP(388*) ratios upon expression by the 2-step and 1-step injection method with or without eRF1(E55D) co-expression. A representative SDS-PAGE gel (prepared as in B) and statistical analysis of data from 6 to 11 experiments including oocytes from 4 to 6 different *X. laevis* frogs per group are shown. Data are represented as mean ± S.D., and significance was determined by a two-tailed unpaired Welch's *t*-test and is indicated as *p<0.05 and ****p<0.0001.

The online version of this article includes the following source data and figure supplement(s) for figure 1:

**Source data 1.** Original gel, *Figure 1B*.

**Source data 2.** Original gel, *Figure 1D*.

**Source data 3.** Original gels for bar graph in *Figure 1D*: Gel2, Gel3, Gel4, and Gel5.

**Source data 4.** Original gels for bar graph in *Figure 1D*: GelA, GelB, and GelC.

**Figure supplement 1.** Variation of experimental conditions to optimize L-3-(6-acetylnaphthalen-2-ylamino)–2-aminopropanoic acid (ANAP) incorporation into oocyte-expressed ion channels.

Figure 1 continued

**Figure supplement 1—source data 1.** Original gel, *Figure 1—figure supplement 1A*.

**Figure supplement 1—source data 2.** Original gel, *Figure 1—figure supplement 1B*.

**Figure supplement 1—source data 3.** Original gel, *Figure 1—figure supplement 1C*.

## Recording of ANAP fluorescence in the head domain reveals mainly gating-associated movements

Next, we recorded ANAP fluorescence changes upon application of 0.3 mM ATP. Control oocytes expressing wt P2X7R showed a gradual fluorescence decrease during ATP application, even when no ANAP was injected (*Figure 3—figure supplement 1*). A similar fluorescence drift was observed with the P2X2R, but not with the hα1 GlyR or the fast-desensitizing P2X1R. The reason for this drift is unclear but needs to be considered when evaluating mutants with small negative fluorescence changes. Specificity of tRNA loading and ANAP-incorporation was evaluated in further control experiments (*Figure 3—figure supplement 2*). Only recordings that met specific inclusion criteria (see Methods) were considered for analysis.

As a prodan derivative, ANAP is highly sensitive to the polarity of its environment and shows a redshift in emission with increasing polarity (*Lee et al., 2009*; *Weber and Farris, 1979*). Consequently, alterations in ANAP fluorescence can be attributed to (i) quenching by ligands, small molecules, or (aromatic) aa, (ii) spectral shift due to changes in the polarity of the environment, or (iii) a combination of these two effects, e.g., in case of ligand interaction.

To allow differentiation between presumably wavelength-independent (de-)quenching of ANAP fluorescence by other molecules or spectral shifts due to changes in the polarity, we simultaneously recorded fluorescence in distinct spectral segments, i.e., (i) 430–470 nm and 470–500 nm with filter set 1 and (ii) 430–490 nm and >500 nm with filter set 2 (*Figure 3A*). This also enabled us to identify mutants that only showed fluorescence changes at certain wavelengths and would have escaped detection, otherwise.

We first recorded ATP-induced fluorescence changes from P2X7Rs containing ANAP in the head domain, which projects over the ATP-binding site (P120-Q128, *Figure 3B*). In agreement with the pronounced conformational changes and ligand interactions of this domain during receptor activation (*Lörinczi et al., 2012*; *McCarthy et al., 2019*), all mutants except E121* and Q128* showed clear fluorescence signals in all spectral segments (*Figure 3C and D*, *Table 1* and *Figure 2—figure supplement 1B*). For E121* and Q128* fluorescence changes were only detected with filter set 2, albeit with minimal changes for E121*. Analysis of mutants P120*, E121*, and P123* with filter set 1 was only preliminary (*Figure 2—figure supplement 1B*) but showed the same trends as signals recorded with filter set 2 (*Figure 3C and D*). Independently of the wavelength, fluorescence changes were always positive for P120* and G126* and negative for Y122*, P123*, and K127*. These consistent changes over the entire ANAP emission spectrum indicate de-/quenching of ANAP either by the ligand ATP and/or other aa residues (see insert table in *Figure 3E*). In contrast, S124* (*Figure 3E*), R125*, and Q128* (*Figure 3C and D* and *Figure 2—figure supplement 1B*) showed positive fluorescence changes in most spectral segments but negative changes for wavelengths >500 nm. The opposite directions imply that these changes result, at least partly, from an ANAP emission shift toward shorter wavelengths and suggest that ANAP enters a less polar environment during receptor activation (*Figure 3E*).

In all head-domain constructs that showed clear kinetics, with the exception of K127*, fluorescence and current changes started simultaneously and showed shorter rising times upon repeated ATP applications, thus recapitulating the characteristic 'current facilitation' of the P2X7R (*Allsopp and Evans, 2015*; *Janks et al., 2019*; *Roger et al., 2008*). To confirm that these fluorescence signals indeed tracked current facilitation, we analyzed the effects of three additional mutants that were expected to affect facilitation: (i) a single point mutation in the juxtamembrane region (S23N) that in human P2X7 was shown to eliminate current facilitation (*Allsopp and Evans, 2015*), (ii) a Cys-Ala mutant (replacement of residues identified to be palmitoylated [*McCarthy et al., 2019*] in the cysteine-rich region by alanine residues [Ser360, Cys362, Cys371, Cys373, Cys374, and Cys377]), and (iii) a ΔCys-mutant (deletion of the cysteine-rich intracellular region, S360-C377 [*McCarthy et al., 2019*; *Roger et al., 2010*]). Contrary to findings in human P2X7 (*Allsopp and Evans, 2015*), the S23N mutation did not eliminate current facilitation in rat P2X7 (*Figure 3—figure supplement 3*). As expected (*Roger et al.,*

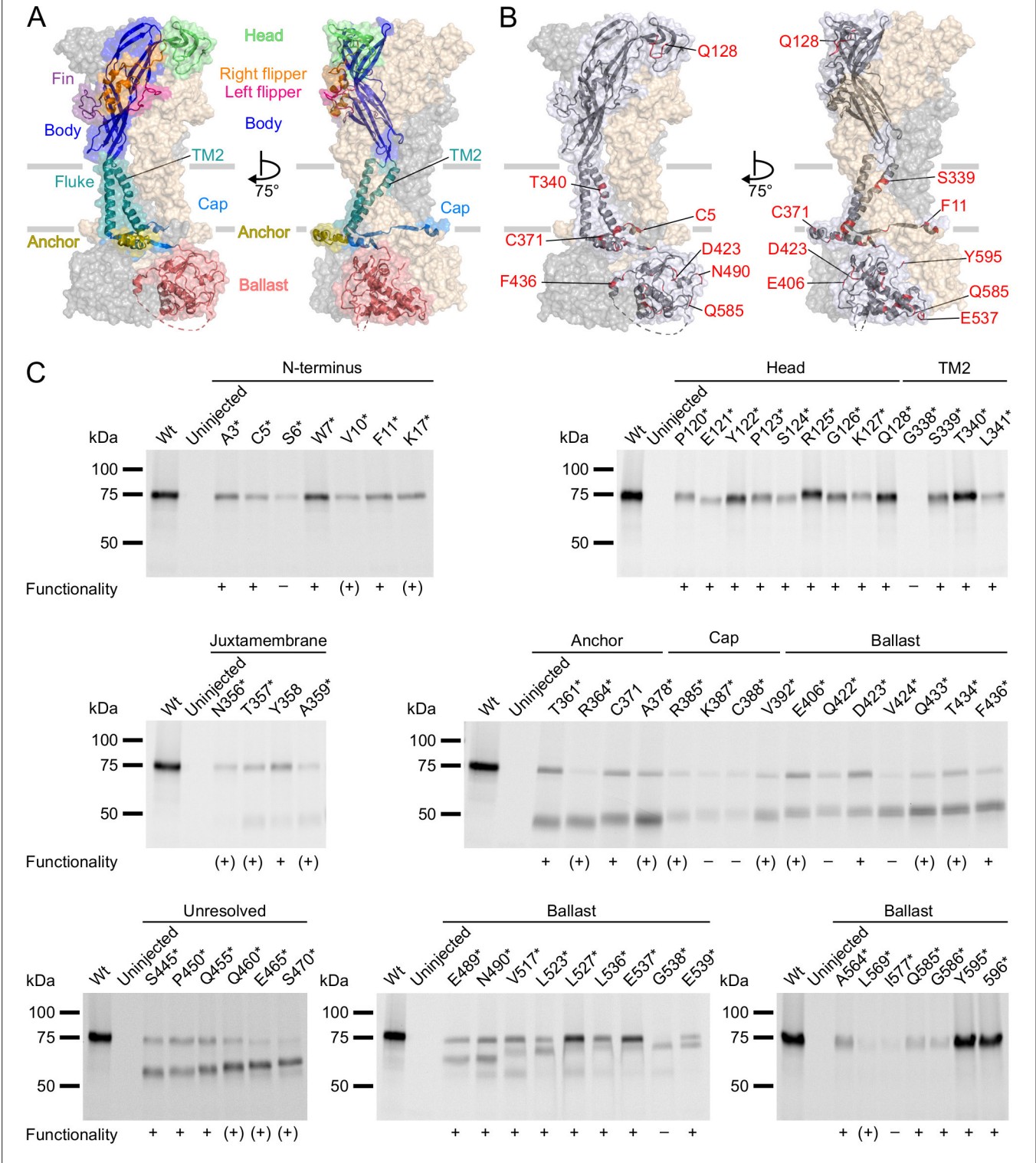

**Figure 2.** P2X7Rs containing L-3-(6-acetylnaphthalen-2-ylamino)–2-aminopropanoic acid (ANAP) at various positions within the extracellular, transmembrane, and cytoplasmic regions are expressed in the plasma membrane and functional. (**A, B**) Surface representations of the rat P2X7 Cryo-EM structure in the open state (PDB ID: 6u9w). The different domains (**A**) and selected sites of ANAP substitutions (**B**) are indicated in one subunit while the two other subunits are shown in gray and wheat, respectively. (**C**) Evaluation of surface expression and functionality of P2X7Rs generated from constructs containing an *amber* stop codon in the indicated positions. *X. laevis* oocytes expressing the constructs were labeled with membrane-impermeant Cy5-NHS ester. His-tagged P2X7Rs were extracted in 0.5% *n*-dodecyl-β-D-maltoside, purified via Ni²⁺ NTA agarose, and analyzed by SDS-PAGE (8%).

*Figure 2 continued*

Symbols indicate current responses to 0.3 mM ATP as determined by two-electrode voltage clamp recordings in the voltage clamp fluorometry setup: +, functional and currents comparable to wt P2X7 after 2–4 days of expression; (+), functional and currents comparable to wt P2X7 after 5–7 days of expression; –, not functional or currents ≤0.5 μA and not reproducible after 4 days. Representative data from two to five independent biochemical experiments are shown.

The online version of this article includes the following source data and figure supplement(s) for figure 2:

**Source data 1.** Original gel (1), *Figure 2C* (N-terminus).

**Source data 2.** Original gel (2), *Figure 2C* (head domain and TM2).

**Source data 3.** Original gel (3), *Figure 2C* (juxtamembrane).

**Source data 4.** Original gel (4), *Figure 2C* (anchor, cap, and ballast domain).

**Source data 5.** Original gel (5), *Figure 2C* (unresolved).

**Source data 6.** Original gel (6), *Figure 2C* (ballast domain, part a).

**Source data 7.** Original gel (7), *Figure 2C* (ballast domain, part b).

**Figure supplement 1.** Comparative analysis of tetramethyl-rhodamine-maleimide (TMRM)-labeled and L-3-(6-acetylnaphthalen-2-ylamino)–2-aminopropanoic acid (ANAP)-labeled P2X7 head domain mutants.

**Figure supplement 1—source data 1.** Original gel, *Figure 2—figure supplement 1A*.

**Figure supplement 1—source data 2.** Summarized data for box plot in *Figure 2—figure supplement 1A*.

**Figure supplement 1—source data 3.** Original recordings for box plot in *Figure 2—figure supplement 1A* and for representative VCF recordings in *Figure 2—figure supplement 1A and B*.

---

*2010*) and in contrast to the wt and most of the analyzed ANAP-containing mutants, current rise times of the ΔCys mutant were not significantly altered between first and second ATP applications, demonstrating that we can indeed identify current facilitation in our setup (*Figure 3—figure supplement 3*). However, the strongly reduced functional expression of the Cys-Ala or ΔCys mutant with additional ANAP-incorporation sites prevented reliable analysis of current and fluorescence kinetics.

Remarkably, the K127* mutant showed a fluorescence change that was faster than the current increase already upon the first receptor activation (*Figure 3F*, *Figure 3—figure supplement 3*). $EC_{50}$ values for ATP were similar at wt, S124* and K127* mutants (*Table 2*). We conclude from this, that ANAP in position 127 reports a process that precedes channel opening and is most likely related to ligand binding, whereas ANAP reports gating-associated conformational changes in the other positions.

## Detection of TM2 movements in response to receptor activation

The following VCF recordings were performed mainly with filter set 2 (430–490 nm and >500 nm), since this revealed more pronounced signals for most mutants.

To exclude fluorescence changes induced by a direct interaction with ATP and to further investigate P2X7 gating, we next investigated positions 339–341 (*Figure 4A*), just preceding S342, the major determinant of the channel gate (*Pippel et al., 2017*). Cysteine substitutions in these positions have previously been shown to be accessible to thiol-reactive dyes only in the open state of the receptor (*Pippel et al., 2017*). In agreement with a critical role in gating, current recordings from mutants S339*, T340*, and L341* were compromised by 10–20-fold higher leak currents compared to wt receptors or other mutants (see also *Figure 4—figure supplement 1*). Nevertheless, they showed clear fluorescence changes during receptor activation (*Figure 4B*), although with higher variability in amplitude and shape between oocytes. While P2X7R mutants S339* and L341* showed positive signals in all spectral ranges, fluorescence changes in T340* were inconsistent at shorter emission wavelengths, but mostly negative below 470 nm, and positive above 470 nm, indicating again that a spectral shift contributed to these signals (*Figure 4B and C*).

Notably, fluorescence signals from P2X7 T340* were also significantly larger during the first ATP application compared to the second (*Figure 4B and C*), suggesting that the environment of this position changed between both ATP applications. An intriguing explanation could be an involvement of this region in the facilitation process. However, as the T340* mutant displayed no change in fluorescence or current kinetics between ATP applications (*Figure 4D* and *Figure 4—figure supplement 1*) the facilitation-associated gating mechanism is likely disturbed by this mutation.

---

**Table 1.** Summary of surface expression, current responses (ΔI), and L-3-(6-acetylnaphthalen-2-ylamino)–2-aminopropanoic acid fluorescence changes (%ΔF/F) of the investigated P2X7 mutants.

| | | Surface expression | | | %ΔF/F Filter set 1 | | %ΔF/F Filter set 2 | |
| | Position | Full-length | Truncated | ΔI | 430–470 nm | 470–500 nm | 430–490 nm | >500 nm |
| --- | --- | --- | --- | --- | --- | --- | --- | --- |
| N-terminus | A3 | + | – | + | ↑ | ↑ | ↑ | ↑ |
| | C5 | + | – | + | (–) | (–) | ↑ | ↑ |
| | S6 | + | – | – | (–) | (–) | (–) | (–) |
| | W7 | + | – | + | (↑) | (↑) | ↑ | ↑ |
| | V10 | + | – | (+) | (–) | (–) | (↑) | (↑) |
| | F11 | + | – | + | (–) | (–) | ↑ | ↑ |
| | K17 | + | – | (+) | (–) | (–) | (–) | (–) |
| Head domain | P120 | + | – | + | (↑) | (↑) | ↑ | ↑ |
| | E121 | + | – | + | (–) | (–) | ↑ | ↑ |
| | Y122 | + | – | + | ↓ | ↓ | ↓ | ↓ |
| | P123 | + | – | + | n.d. | n.d. | ↓ | ↓ |
| | S124 | + | – | + | ↑ | ↑ | ↑ | ↓ |
| | R125 | + | – | + | ↑ | ↑ | ↑ | ↓ |
| | G126 | + | – | + | ↑ | ↑ | ↑ | ↑ |
| | K127 | + | – | + | ↓ | ↓ | ↓ | ↓ |
| | Q128 | + | – | + | – | – | ↑ | ↓ |
| TM2 | G338 | – | – | – | n.d. | n.d. | (–) | (–) |
| | S339 | + | – | + | ↑ | ↑ | ↑ | ↑ |
| | T340 | + | – | + | – / ↓ | ↑ | ↑ | ↑ |
| | L341 | + | – | + | (↑) | (↑) | ↑ | ↑ |

*Table 1 continued on next page*

*Table 1 continued*

| | Position | Surface expression | | | %ΔF/F Filter set 1 | | %ΔF/F Filter set 2 | |
|---|---|---|---|---|---|---|---|---|
| | | Full-length | Truncated | ΔI | 430–470 nm | 470–500 nm | 430–490 nm | >500 nm |
| | N356 | + | – | (+) | (–) | (–) | (–) | (–) |
| | T357 | + | + | (+) | (–) | (–) | (–) | (–) |
| | Y358 | + | + | + | (–) | (–) | – | – |
| | A359 | + | + | (+) | (–) | (–) | (–) | (–) |
| | T361 | + | + | + | (–) | (↓) | – | ↓ |
| | R364 | + | + | (+) | n.d. | n.d. | (–) | (–) |
| | C371 | + | + | + | – | – | (–) | (–) |
| | A378 | + | + | (+) | (–) | (–) | n.d. | n.d. |
| | R385 | + | + | (+) | n.d. | n.d. | (–) | (–) |
| C-terminus | K387 | + | + | – | n.d. | n.d. | (–) | (–) |
| | C388 | + | + | – | n.d. | n.d. | (–) | (–) |
| | V392 | + | + | (+) | (–) | (–) | (–) | (–) |
| | E406 | + | + | (+) | (–) | (–) | (–) | (–) |
| | Q422 | + | + | – | n.d. | n.d. | (–) | (–) |
| | D423 | + | + | + | (–) | (–) | ↑ | ↑ |
| | V424 | + | + | – | n.d. | n.d. | (–) | (–) |
| | Q433 | + | + | (+) | (–) | (–) | n.d. | n.d. |
| | T434 | + | + | (+) | (–) | (–) | n.d. | n.d. |
| | F436 | + | + | + | n.d. | n.d. | (–) | (–) |
| Unresolved | S445 | + | + | + | (–) | (–) | – | (–) |
| | P450 | + | + | + | – | – | – | – |
| | Q455 | + | + | + | – | – | – | – |
| | Q460 | + | + | (+) | (–) | (–) | (–) | (–) |
| | E465 | + | + | (+) | (–) | (–) | n.d. | n.d. |
| | S470 | + | + | (+) | (–) | (–) | n.d. | n.d. |

*Table 1 continued on next page*

*Table 1 continued*

| Position | Surface expression | | | %ΔF/F Filter set 1 | | %ΔF/F Filter set 2 | |
|---|---|---|---|---|---|---|---|
| | Full-length | Truncated | ΔI | 430–470 nm | 470–500 nm | 430–490 nm | >500 nm |
| E489 | + | + | + | (–) | (–) | – | – |
| N490 | + | + | + | (–) | (–) | – | – |
| V517 | + | + | + | n.d. | n.d. | – | – |
| L523 | + | + | + | n.d. | n.d. | – | – |
| L527 | + | + | + | n.d. | n.d. | – | – |
| L536 | + | + | + | (–) | (–) | – | – |
| E537 | + | + | + | – | – | – | – |
| G538 | – | + | – | (–) | (–) | (–) | (–) |
| E539 | + | + | + | (–) | (–) | – | – |
| A564 | + | ? | + | n.d. | n.d. | ↑ | ↑ |
| L569 | + | ? | (+) | n.d. | n.d. | (–) | (–) |
| I577 | + | ? | – | n.d. | n.d. | (–) | (–) |
| Q585 | + | ? | + | (–) | (–) | – | – |
| G586 | + | ? | + | (–) | (–) | – | – |
| Y595 | + | ? | + | – | – | – | – |
| 596 | + | ? | + | – | – | – | – |

(C-terminus applies to the rows above)

+ and – indicate presence and absence of protein or signals, respectively. In case of current responses, + means response comparable to wt receptors and (+) means reduced responses. ↑ and ↓ indicate positive and negative fluorescence signals, respectively. 3–50 oocytes were measured per construct and filter set. In case of fluorescence responses, symbols in brackets indicate where less than three recordings met the criteria defined in the methods (mostly because of impaired fucntionality) and represent tendencies only. ?, not distinguishable (because of similar length of full-length and truncated constructs); n.d., not determined.

The online version of this article includes the following source data for table 1:

**Source data 1.** Summarized data for *Table 1* with assignment to the original VCF recordings; also including data from *Figure 2—figure supplement 1B* (box plot); *Figure 3C, D, E, F*; *Figure 3—figure supplement 3C*; *Figure 4B, C, D*; *Figure 4—figure supplement 1B*; and *Figure 5B, C, D*. The respective original recordings are deposited with Dryad.

## Scanning of the P2X7 intracellular domains for ATP-induced conformational changes

The large intracellular P2X7 C-terminus mediates many of the P2X7R downstream effects (*Kopp et al., 2019*). While the P2X7 cryo-EM structures revealed the role of the juxtamembrane N- and C-terminal domains in receptor desensitization, their role in downstream signaling and in particular the molecular function of the ballast domain remain completely unclear. Analysis of ANAP fluorescence changes within the cytoplasmic domain was therefore a primary aim of this study. We first introduced ANAP into juxtamembrane regions within the N- and C-termini (*Figure 5A*) that form the cytoplasmic cap and anchor domains excluding palmitoylated residues (C4, S360, C362, C363, C374, and C377) (*McCarthy et al., 2019*). Although all receptors with N-terminal ANAP substitutions were formed in full length (*Figure 2C*), current and fluorescence responses for S6*, V10*, and K17* substitutions remained small and inconsistent even after 6 days of expression. A3*, C5*, W7*, and F11* mutants showed positive fluorescence signals of variable sizes (*Figure 5B*), and the kinetics of F11* fluorescence correlated with current facilitation (*Figure 5C*, *Figure 3—figure supplement 3*). ANAP fluorescence in F11* was not quenched by the nearby Trp (W7), as its removal had no apparent effect (*Figure 5—figure supplement 1*). Within the juxtamembrane C-terminal regions,

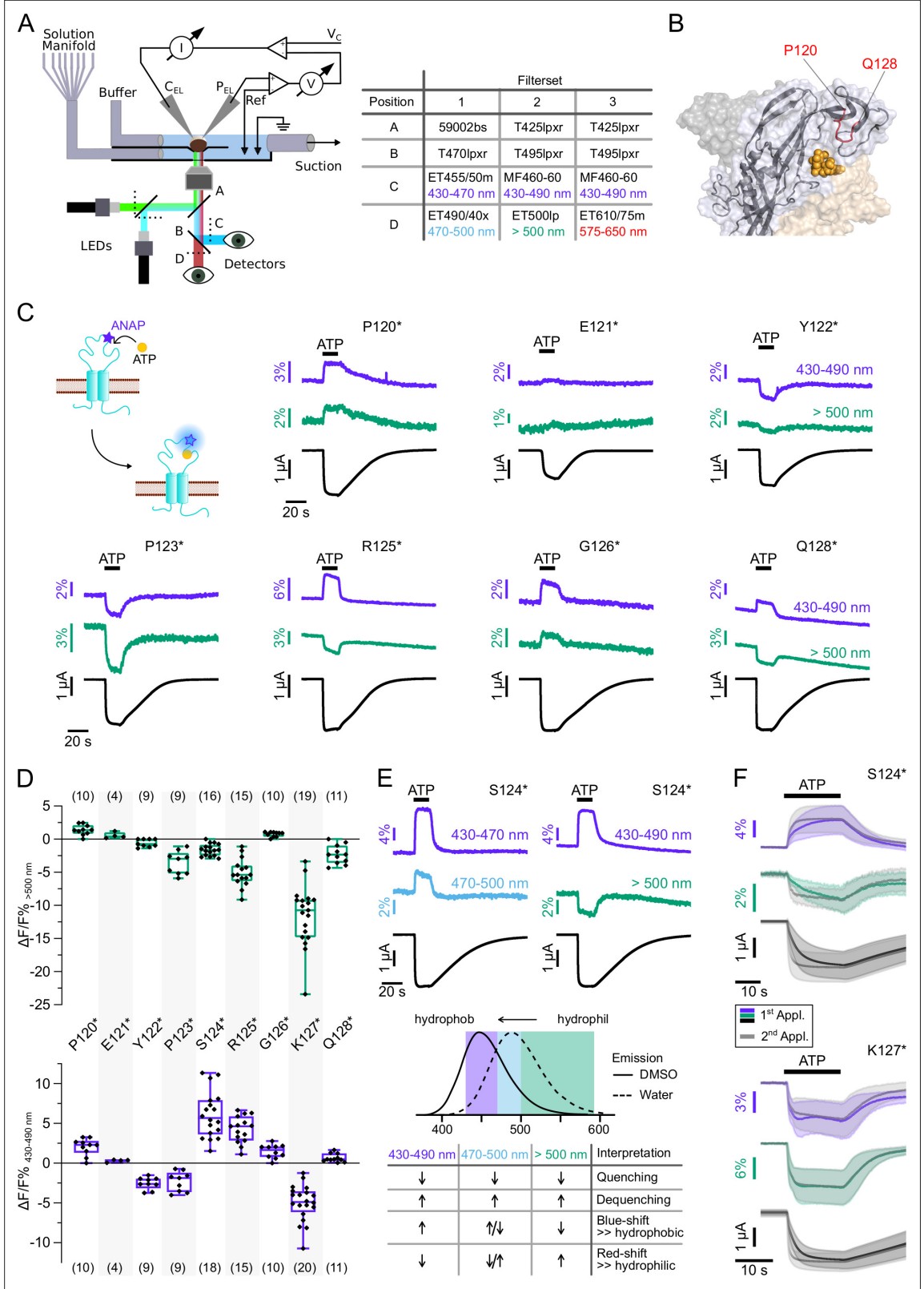

**Figure 3.** Characterization of ATP-induced fluorescence changes in the P2X7 head domain recorded at different wavelengths. (**A**) Schematic of the voltage clamp fluorometry (VCF)-recording system and summary of filter/dichroic mirror configurations used to detect distinct spectral parts of L-3-(6-acetylnaphthalen-2-ylamino)–2-aminopropanoic acid (ANAP)-fluorescence (sets 1 and 2) and ANAP in combination with tetramethyl-rhodamine-maleimide (TMRM) or R-GECO1.2 (set 3). The corresponding positions A, B, C, and D are shown in the schematic. A second LED (green) was used for

*Figure 3 continued on next page*

*Figure 3 continued*

additional excitation of TMRM or R-GECO1.2. (**B**) Close-up of the P2X7 head domain in surface representation indicating the ANAP-substituted amino acid residues P120-Q128 (red). The three subunits are colored in gray, wheat, and light blue. (**C**) Principle of VCF and representative VCF recordings in response to 0.3 mM ATP (upon second application). Change of fluorescence intensity of a site-specifically introduced environment-sensitive fluorophore can be induced by ligand binding and/or conformational changes. (**D**) Box plots summarizing results from the indicated ANAP-labeled P2X7Rs at two different emission wavelengths with ΔF/F% representing the maximum fluorescence signal during a 15-s ATP application. Numbers of recordings are given in brackets. (**E**) Representative VCF recordings in response to 0.3 mM ATP of P2X7(S124*) at three different emission wavelengths and summary of most likely interpretations. Note that fluorescence changes are most likely resulting from multiple effects, and only the dominant effect is stated. Arrows indicate direction of fluorescent changes. (**F**) Overlay of VCF recordings upon first (colored) and second (gray) ATP applications (0.3 mM) at two different emission wavelengths for P2X7(S124*) (14 oocytes) and P2X7(K127*) (17 oocytes), respectively. Averaged VCF recordings are shown as lines, and standard deviations are plotted as envelopes. Baseline currents (15 s before ATP application) were adjusted for clarity. All recordings were performed in divalent-free buffer, and oocytes were clamped at –30 mV. Original recordings have also been deposited with Dryad and summarized and assigned in *Table 1—source data 1*.

The online version of this article includes the following source data and figure supplement(s) for figure 3:

**Source data 1.** Original recordings, *Figure 3C, E and F*.

**Figure supplement 1.** Control voltage clamp fluorometry (VCF) recordings from oocytes expressing different non-mutated ion channels.

**Figure supplement 1—source data 1.** Original recordings, *Figure 3—figure supplement 1*.

**Figure supplement 2.** Control experiments to test the specificity of tRNA-loading and L-3-(6-acetylnaphthalen-2-ylamino)–2-aminopropanoic acid (ANAP) incorporation into P2X7.

**Figure supplement 2—source data 1.** Original recordings, *Figure 3—figure supplement 2A*.

**Figure supplement 2—source data 2.** Original gel, *Figure 3—figure supplement 2B*.

**Figure supplement 3.** Deletion of the cysteine-rich region eliminates current facilitation, and F11* and S124* mutants track current facilitation.

**Figure supplement 3—source data 1.** Summarized data, *Figure 3—figure supplement 3A and B*.

**Figure supplement 3—source data 2.** Original recordings, *Figure 3—figure supplement 3A and B*.

**Figure supplement 3—source data 3.** Original recordings, *Figure 3—figure supplement 3C*.

ANAP was introduced between TM2 and the anchor domain (N356*, T357*, Y358*, A359*), upstream of $\beta_{15}$, which is part of the cytoplasmic cap structure (T361*, R364*, C371*, A378*, R385*, K387*), and upstream of the cytosolic ballast domain (C388* and V392*). Surface expression of functional full-length receptors was observed for all constructs except for K387* and C388*. In contrast to the juxtamembrane N-terminal residues, however, only one of these C-terminal mutants, T361*, showed a fluorescence change, albeit in only ~50% of the recordings (*Figure 5D*). Interestingly, both F11 and T361 lie within two of at least four possible cholesterol recognition amino acid consensus (CRAC)

**Table 2.** EC$_{50}$ values for ATP and Hill coefficients (n$_H$) at wt and L-3-(6-acetylnaphthalen-2-ylamino)–2-aminopropanoic acid-containing P2X7 receptor constructs.

| Mutant | EC$_{50}$ (M) | n$_H$ |
|---|---|---|
| Wt | 4.202e-005 (3.211e-005–5.704e-005) | 1.049 (0.7962–1.380) |
| F11* | 7.802e-005 (6.268e-005–9.893e-005) | 1.148 (0.9345–1.410) |
| S124* | 8.316e-005 (6.068e-005–0.0001236) | 1.122 (0.8271–1.519) |
| F11*, S124C | 0.0001003 (8.439e-005–0.0001216) | 1.290 (1.069–1.571) |
| K127* | 6.511e-005 (4.281e-005–0.0001339) | 0.6601 (0.4779–0.8662) |
| D423* | 6.513e-005 (4.729e-005–0.0001057) | 1.240 (0.8087–1.821) |
| A564* | 5.159e-005 (3.491e-005–9.976e-005) | 0.7810 (0.5364–1.087) |

Number in brackets are 95% confidence intervals, n=3–11.

The online version of this article includes the following source data for table 2:

**Source data 1.** Original recordings for *Table 2*; *Figure 3—figure supplement 3D*; and *Figure 5—figure supplement 2A*.

**Source data 2.** Summarized data for *Table 2*; *Figure 3—figure supplement 3D*; and *Figure 5—figure supplement 2A*.

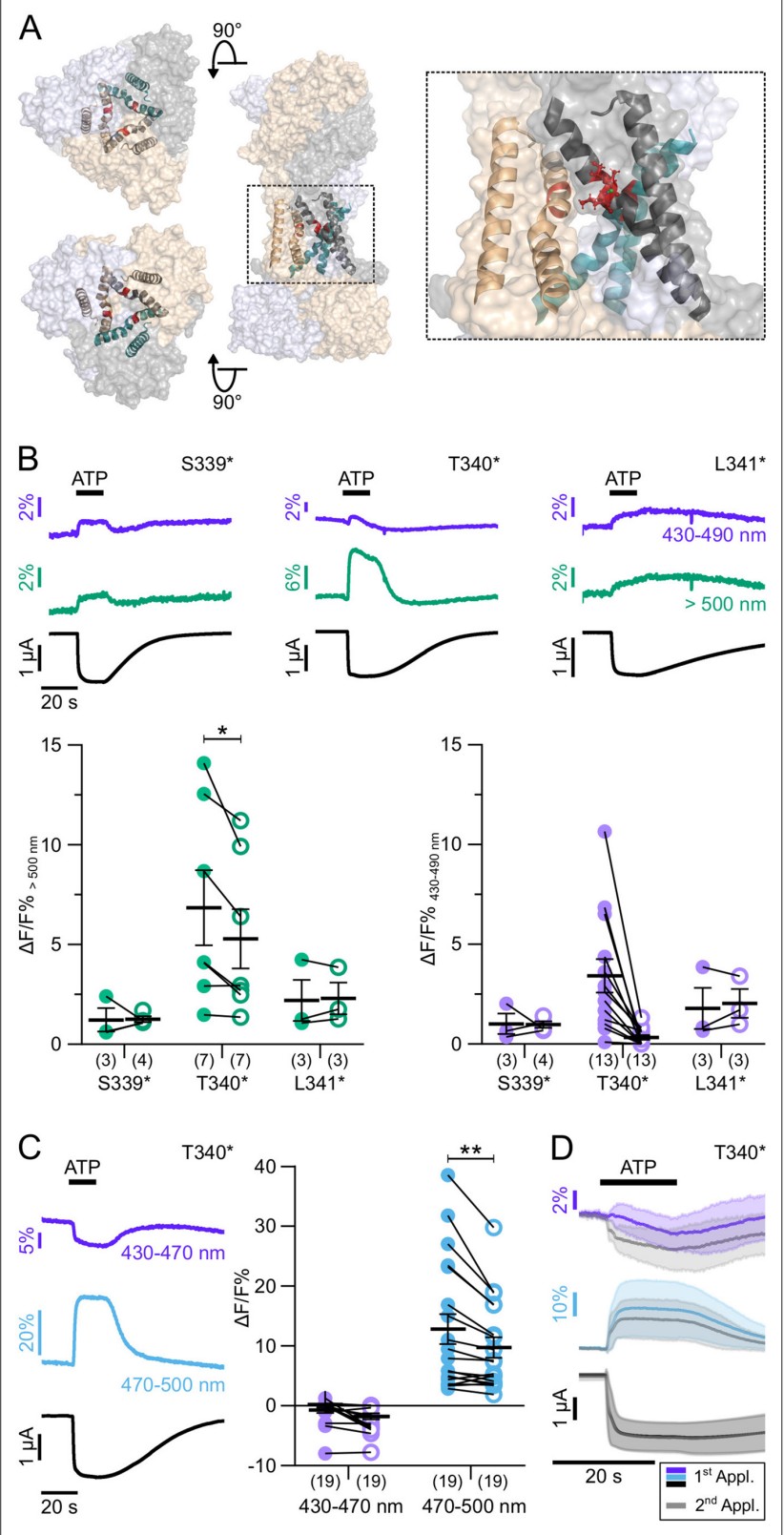

**Figure 4.** ATP-induced L-3-(6-acetylnaphthalen-2-ylamino)–2-aminopropanoic acid (ANAP) fluorescence changes in the P2X7 TM2 domain. (**A**) Overview and close-up of the three P2X7 subunits (in wheat, gray, and purple) with the TM helices as cartoon representations (in wheat, gray, and green) and the ANAP-substituted residues S339, T340, and L341 (in red). (**B**) Representative voltage clamp fluorometry (VCF) recordings from the indicated

*Figure 4 continued on next page*

*Figure 4 continued*

mutants in response to 0.3 mM ATP (upon second application) and summary of results at two different emission wavelengths. Note that recordings from all constructs were compromised by high leak currents. Graphs compare maximal fluorescence signals during first (closed circles) and second (open circles) ATP applications (interval 195 s). Data are represented as mean ± S.E.M. Significance was determined using the two-tailed paired Student's *t*-test (*, p<0.05; **, p<0.005). (**C**) Representative recordings and summary (performed as in B) from P2X7(T340*) with filter set 2. (**D**) Overlay of VCF recordings from P2X7(T340*) upon first (colored) and second (gray) ATP applications (0.3 mM) at two different emission wavelengths. Averaged VCF recordings from 11 oocytes are shown as lines, and standard deviations are plotted as envelopes. Baseline currents (15 s before ATP application) were adjusted for clarity. All recordings were performed in divalent-free buffer, and oocytes were clamped at –30 mV. Wavelengths passed by the used filter sets are indicated. Original recordings have also been deposited with Dryad and summarized and assigned in *Table 1—source data 1*.

The online version of this article includes the following source data and figure supplement(s) for figure 4:

**Source data 1.** Original recordings, *Figure 4B, C, and D*.

**Figure supplement 1.** L-3-(6-acetylnaphthalen-2-ylamino)–2-aminopropanoic acid (ANAP) in TM2 causes leakiness and affects current facilitation.

**Figure supplement 1—source data 1.** Original recordings, *Figure 4—figure supplement 1A*.

motifs that have been proposed to be involved in the cholesterol sensitivity of P2X7 channel gating (*Robinson et al., 2014*).

ANAP introduction in most of the 29 ballast domain positions led to a dominant formation of truncated protein, indicating that this domain does not tolerate substitutions very well and/or that the truncated constructs form stable proteins. Four of these mutants (Q422*, V424*, G538*, I577*) did not form functional receptors at all. For most of the remaining constructs, no specific fluorescence changes could be detected, despite promising surface transport and current responses comparable to wt receptors for at least 12 of them (see *Table 1*, *Figure 2C*).

Only in two mutants, A564* and D423*, fluorescence changes could be recorded: A564* showed clearly positive signals, while D423* showed positive signals in only ~40% of the recordings (*Figure 5D*). Both mutants showed EC$_{50}$ values similar to wt P2X7 and were not functional in control oocytes injected without ANAP (*Figure 5—figure supplement 2*), suggesting that the respective truncated proteins (compare *Figure 2C*) do either not contribute to current responses or only in complex with full-length (ANAP-containing) P2X7 subunits. D423 is located within a loop connecting the $\beta_{17}$ and $\beta_{18}$ strands and situated on the outer surface of the cytoplasmic ballast, facing away from both the central axis of the receptor and the neighboring subunits (*Figure 5F*). Notably, mutation of the neighboring positions (Q422*, V424*) resulted in non-functional receptors. A564 is located in the $\alpha_{15}$ helix at the very end of a cavity formed by the $\alpha_{13}$, $\alpha_{14}$, and $\alpha_{16}$ helices and a short $\alpha_9$ helix of the neighboring subunit (*Figure 5E*). This cavity harbors the guanosine nucleotide binding site identified by cryo-EM and liquid chromatography-tandem mass spectrometry analysis, and GDP was found to interact with residues A567 and L569 (*McCarthy et al., 2019*), both in close proximity to A564. $\alpha_{16}$, is also part of a proposed lipid interaction or lipopolysaccharide (LPS) binding motif (*Denlinger et al., 2001*) and $\alpha_{14}$ at the bottom of the cavity is part of a proposed calcium-dependent calmodulin binding motif (residues I541-S560) (*Roger et al., 2010*; *Roger et al., 2008*). To identify possible palmitoylation or CaM-dependent movements of the ballast domain or effects on receptor function, we analyzed the influence of the non-palmitoylated ΔCys- and Cys-Ala mutants (*Figure 3—figure supplement 3*) as well as a ΔCaM mutant, in which a proposed calmodulin binding site was deleted (*Roger et al., 2010*) on ANAP fluorescence. While the poor expression of the ΔCys and Cys-Ala mutants in combination with ANAP prevented VCF analysis, combination of the ΔCaM mutation with ANAP (in intracellular positions F11*, D423*, or A564* or in the head domain S124*, K127*) yielded good expression and similar current kinetics and fluorescence changes, as observed before for the single mutants (*Figure 5—figure supplement 3*). This argues against a major functional effect of the CaM binding site mutation on the current facilitation or on molecular movements, at least in the oocyte-expressed receptor.

Taken together, only two positions, D423 and A564, could be identified within the ballast domain, where ANAP reported environmental changes, suggesting only limited ATP-induced movements in

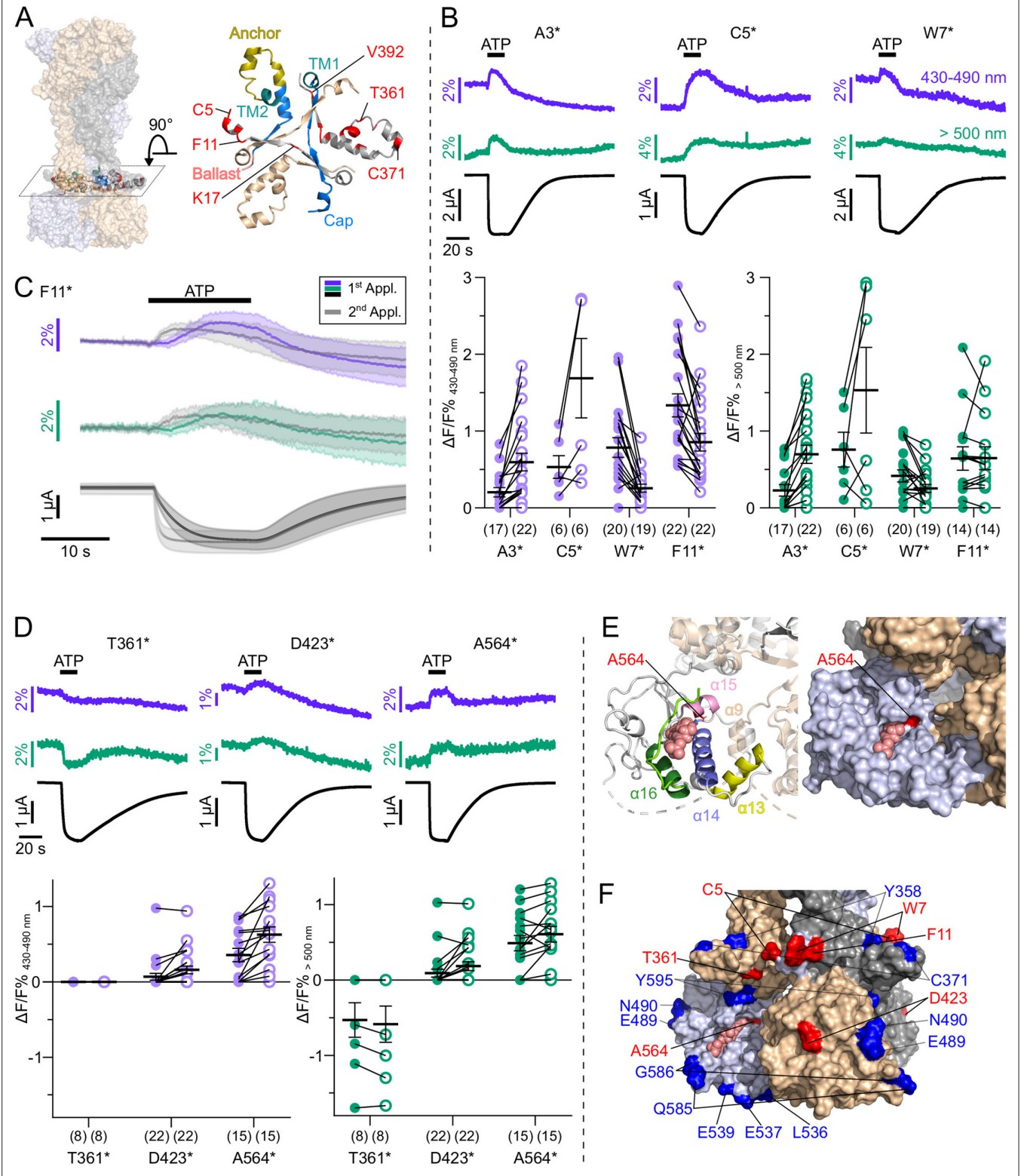

**Figure 5.** L-3-(6-acetylnaphthalen-2-ylamino)–2-aminopropanoic acid (ANAP) incorporation into 41 positions of the cytosolic P2X7 domain identified seven positions that report ATP-induced fluorescence changes. (**A**) Surface representation of all three P2X7 subunits (in wheat, gray, and purple) showing location of the juxtamembrane regions and close up (top view) detailing the anchor and cap domains (in yellow and blue, respectively) and ANAP-substituted positions (in red) within a single P2X7 subunit. (**B**) Representative voltage clamp fluorometry (VCF) recordings and data summary from P2X7R mutants containing ANAP at different positions within the N-terminus. Responses to 0.3 mM ATP were recorded at two different emission

*Figure 5 continued on next page*

*Figure 5 continued*

wavelengths. Graphs compare maximal fluorescence signals during first (closed circles) and second (open circles) ATP applications (interval 195 s). Data are represented as mean ± S.E.M. (**C**) Overlay of VCF recordings from P2X7(F11*) upon first (colored) and second (gray) ATP application (0.3 mM) at two different emission wavelengths. Lines represent averaged VCF recordings from 13 oocytes. Standard deviations are plotted as envelopes. Baseline currents (15 s before ATP application) were adjusted for clarity. (**D**) Representative VCF recordings from the indicated mutants in response to a second application of 0.3 mM ATP and summary of results at the indicated emission wavelengths (performed as in B). Graphs compare maximal fluorescence signals during first (closed circles) and second (open circles) ATP applications (interval 195 s). Data are represented as mean ± S.E.M. All recordings were performed in divalent-free buffer, and oocytes were clamped at –30 mV. (**E**) Close-up of the cytoplasmic ballast domain from one P2X7 subunit in cartoon and surface representation highlighting a bound GDP (salmon), surrounding α-helices, and residue A564 (red). (**F**) Surface representation of the cytoplasmic domains of all three P2X7 subunits (in gray, light blue, and wheat) with bound GDP (salmon). Positions in which ATP-induced ANAP fluorescence changes were identified are shown in red. ANAP-substituted positions in which no fluorescence changes were seen (despite surface expression and current responses) are shown in blue. Original recordings have also been deposited with Dryad and summarized and assigned in *Table 1—source data 1*.

The online version of this article includes the following source data and figure supplement(s) for figure 5:

**Source data 1.** Original recordings, *Figure 5B and D*.

**Figure supplement 1.** The fluorescence change in P2X7(F11*) is not caused by a dequenching effect of the nearby Trp residue.

**Figure supplement 1—source data 1.** Summarized data for bar graph in *Figure 5—figure supplement 1B*.

**Figure supplement 1—source data 2.** Original recordings, *Figure 5—figure supplement 1A and B*.

**Figure supplement 1—source data 3.** Original gel, *Figure 5—figure supplement 1C*.

**Figure supplement 2.** Dose-response analysis for intracellular P2X7 mutants F11*, D423*, and A564* and contribution of D423 and A564 deletion mutants to current responses.

**Figure supplement 2—source data 1.** Original recordings, *Figure 5—figure supplement 2B*.

**Figure supplement 2—source data 2.** Summarized data, *Figure 5—figure supplement 2B*.

**Figure supplement 3.** Elimination of a CaM-binding motif has no apparent effect on current kinetics or fluorescence responses.

**Figure supplement 3—source data 1.** Summarized data for box plot in *Figure 5—figure supplement 3A* and representative VCF recordings in *Figure 5—figure supplement 3B*.

**Figure supplement 3—source data 2.** Original recordings, *Figure 5—figure supplement 3A and B*.

this domain. However, mutant A564* has great potential as a reporter for yet undefined processes that affect GDP binding and/or metabolism.

## Parallel recording of ANAP fluorescence with other fluorophores

Based on the above findings, we propose that yet unknown intracellular ligands or protein inter-actors are required to mediate downstream signaling via the ballast domain. As potential tools to further investigate such molecules and the dynamics of their molecular interplay with the P2X7R, we combined ANAP with other fluorophores and equipped the VCF setup with a second LED for parallel excitation of two different fluorophores within the same protein.

First, we generated a double mutant (F11*/ S124C) suited to investigate the dynamics of P2X7 activation in different parts of the receptor by parallel labeling with the thiol-reactive fluorophore TMRM in the extracellular head domain and with ANAP in the cytoplasmic N-terminus. As seen in *Figure 6A*, and similar to ANAP in K127*, TMRM in the head-domain showed an instant fluorescent change already upon a first ATP application, whereas the ANAP fluorescence change in F11* was clearly slower. However, both signals coincided upon a second ATP application, further supporting our hypothesis that the so-called current facilitation in P2X7 is due to a change in receptor gating rather than ligand binding. $EC_{50}$ values for ATP at these mutants and at wt P2X7 were comparable (*Figure 3—figure supplement 3* and *Table 2*).

Since P2X7 is known to permeate $Ca^{2+}$, an important mediator of intracellular signaling, we also established a protocol to combine VCF recording of ANAP-fluorescence with imaging of P2X7-mediated $Ca^{2+}$ influx by fusing the genetically encoded $Ca^{2+}$-sensor R-GECO1.2 (*Wu et al., 2013*) C-terminally to the receptor. Combination of P2X7 R-GECO1.2 with the K127* mutant, in which ANAP most likely reports a ligand binding-associated process, showed a clearly delayed onset of the $Ca^{2+}$-dependent R-GECO fluorescence signal, as expected (*Figure 6B*). A limitation of this protocol was, however, that $Ca^{2+}$ promoted P2X7 desensitization and affected baseline fluorescence, specifically

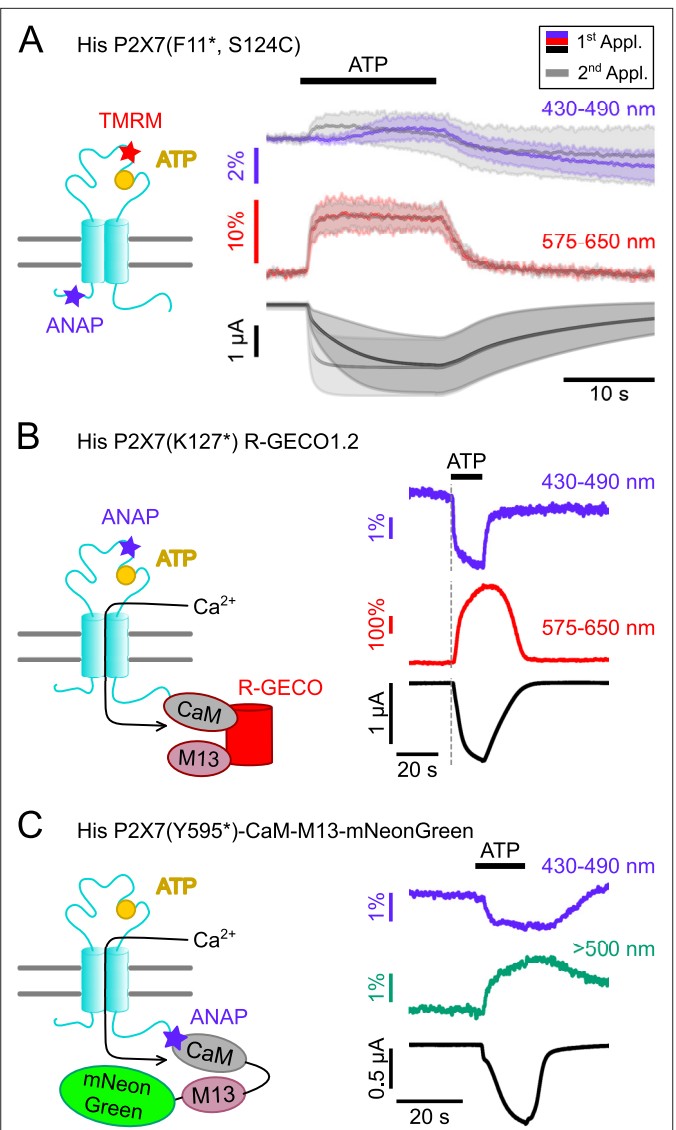

**Figure 6.** Double-labeled P2X7Rs as potential tools to analyze intracellular domain movements, downstream signaling events, and protein interactions. (**A**) Scheme of a P2X7 subunit double-labeled with L-3-(6-acetylnaphthalen-2-ylamino)–2-aminopropanoic acid (ANAP) and tetramethyl-rhodamine-maleimide (TMRM) (F11*, S124C) and overlay of fluorescence and current responses to first (colored) and second (gray) ATP applications (0.3 mM) at the indicated emission wavelengths. Lines represent averaged voltage clamp fluorometry (VCF) recordings from five different oocytes and standard deviations are plotted as envelopes. Baseline currents (15 s before ATP application) were adjusted for clarity. (**B**) Scheme of P2X7(K127*) subunit C-terminally fused to R-GECO1.2 and representative VCF recording in response to 0.3 mM ATP. Recordings were performed in buffer containing 0.5 mM $Ca^{2+}$. (**C**) Scheme showing the P2X7(Y595*)-CaM-M13-mNeonGreen construct that served as positive control for recordings of FRET between ANAP and mNeonGreen. $Ca^{2+}$ entry through the P2X7R is supposed to induce conformational changes in the CaM-M13-mNeonGreen reporter, which are detected as a FRET signal. A representative VCF recording in response to 0.3 mM ATP is shown. In all recordings, oocytes were clamped at –30 mV.

The online version of this article includes the following source data and figure supplement(s) for figure 6:

**Source data 1.** Original recordings, *Figure 6A, B, and C*.

**Figure supplement 1.** $Ca^{2+}$-containing buffers cause large fluorescence changes, even in the absence of L-3-(6-acetylnaphthalen-2-ylamino)–2-aminopropanoic acid (ANAP).

**Figure supplement 1—source data 1.** Original recordings, *Figure 6—figure supplement 1A, B, C*.

**Figure supplement 1—source data 2.** Original and summarized photometric data for *Figure 6—figure*

*Figure 6 continued*

*supplement 1E*.

**Figure supplement 2.** Control constructs and corresponding voltage clamp fluorometry recordings to confirm the specificity of the FRET signals.

**Figure supplement 2—source data 1.** Summarized data for bar graph in *Figure 6—figure supplement 2*.

**Figure supplement 2—source data 2.** Original recordings for bar graph in *Figure 6—figure supplement 2*.

**Figure supplement 3.** Experiments with L-3-(6-acetylnaphthalen-2-ylamino)–2-aminopropanoic acid (ANAP)-containing P2X7 constructs and soluble mNeonGreen-tagged CaM reveal unspecific fluorescence signals.

**Figure supplement 3—source data 1.** Summarized data for bar graph in *Figure 6—figure supplement 3*.

**Figure supplement 3—source data 2.** Original recordings for bar graph in *Figure 6—figure supplement 3*.

in the ANAP emission spectrum (*Figure 6—figure supplement 1*). Use of an alternative fluorescent unnatural amino acid (fUAA) would therefore be advantageous.

ANAP has been successfully used as a FRET partner in combination with acceptor transition metals (*Gordon et al., 2018*), with EGFP (*Mitchell et al., 2017*), and with YFP to study the apoptosis-regulating Bax-Hsp70 interaction in HeLa cells (*Park et al., 2019*) and the interaction between BACE1 and KCNQ2/3 in tsA-201 cells (*Dai, 2022*). Thus, we finally tested whether we could detect FRET signals between ANAP and potential interactors carrying a mNeonGreen-tag. As a proof of concept and based on a CaM-M13-EGFP fusion protein (*Mitchell et al., 2017*), we generated a positive control (P2X7(Y595*)CaM-M13-mNeonGreen), in which ANAP was introduced into the very C-terminus of a P2X7R that was C-terminally fused to a construct consisting of calmodulin (CaM), CaM-binding myosin light chain kinase (M13), and mNeonGreen (*Shaner et al., 2013*). Upon $Ca^{2+}$-binding, this construct should move the acceptor protein mNeonGreen in closer proximity to ANAP, which acts as FRET donor.

As expected, ATP-induced $Ca^{2+}$-influx reduced ANAP fluorescence and increased mNeon-Green fluorescence (*Figure 6C*). The specificity of the signals was confirmed in control experiments (*Figure 6—figure supplement 2*).

Driven by these results we sought to investigate a potential interaction between the rat P2X7 receptor and CaM (*Roger et al., 2010*; *Roger et al., 2008*) and performed experiments with ANAP-labeled P2X7 receptors and mNeonGreen-tagged CaM. However, these recordings revealed no differences to the negative controls, as the CaM-mNeonGreen construct yielded unspecific fluorescence signals (*Figure 6—figure supplement 3*), possibly due to interaction of soluble mNeonGreen-tagged CaM with the co-injected ANAP.

Since the small FRET signals additionally complicated these analyses, the use of another fUAA with superior photophysical properties such as Acd (*Zagotta et al., 2021*) might provide a better alternative.

In summary, we identified kinetically different fluorescence changes in the head domain that are most likely associated with ligand binding and gating, respectively, and suggest an involvement of the region around T340 in P2X7 current facilitation. We find, however, only limited ATP-induced movements in the intracellular domains and hypothesize that additional interactions might be required to 'activate' the ballast domain. Protocols for parallel recordings of ANAP with TMRM, mNeonGreen, and R-Geco1.2 were established to further analyze such interactions.

## Discussion
### Optimization of UAA incorporation into P2X7

Site-specific UAA-incorporation represents a powerful method for protein structure-function analysis, and protocols exist for several model systems (*Braun et al., 2020*; *Klippenstein et al., 2018*; *Leisle et al., 2015*; *Pless et al., 2015*). In *X. laevis* oocytes, stop codon suppression either by *in vitro* synthesized UAA-aminoacylated tRNAs or by expression of co-evolved tRNA/aminoacyl-tRNA synthetase pairs has been established. Recently, the semisynthetic ligation of peptide fragments containing the modification using split intervening proteins (inteins) (*Sarkar et al., 2021*) has also been described (*Galleano et al., 2021*; *Khoo et al., 2020*). While chemically aminoacylated tRNA cannot be reloaded after deacylation without a tRNA synthetase (*Klippenstein et al., 2018*), expression of co-evolved

orthogonal tRNA/aminoacyl-tRNA synthetase pairs requires an additional nuclear injection (*Kalstrup and Blunck, 2013*; *Ye et al., 2013*). Here, we combined both methods by simultaneously injecting a synthesized suppressor tRNA, cRNA encoding the tRNA synthetase, ANAP, and cRNA encoding the target protein into the cytoplasm. We further enhanced ANAP incorporation by co-injection of cRNA encoding mutated *X. laevis* eRF1, disfavoring premature translation termination. While mutated eRF1 could potentially interfere with correct translation of endogenous *amber*-terminated oocyte proteins, we observed no apparent impact on oocyte properties. The presented procedure also improved oocyte quality, expression efficiency, and reproducibility and facilitated optimization of injection ratios. While it does not require equipment for synthesis and purification of UAA-labeled tRNA and is easily applicable in a molecular biology lab, it still depends on a co-evolved tRNA/aminoacyl-tRNA synthetase pair. In combination with UAAs suitable for click chemistry, its flexibility and the choice of fluorophores or functional groups could be greatly expanded (*Braun et al., 2020*). Here, we could successfully employ the optimized ANAP labeling strategy to explore conformational changes associated with P2X7R activation.

## Is P2X7 current facilitation an intrinsic receptor property?

Based on crystal and cryo-EM structures, a molecular mechanism of P2XR gating has been established: ATP-binding to its extracellular inter-subunit binding sites leads to a jaw-like tightening of the head and dorsal fin domains of neighboring subunits around the ATP molecule. This induces an upward movement of β strands in the lower part of the extracellular domain and associated pore opening. Upon prolonged and/or repeated activation, the P2X7R shows a characteristic increase in current amplitude and speed of channel opening, which is generally associated with a shift toward higher ATP sensitivity. Several mechanisms have been proposed to contribute to this so-called current facilitation: modulation of receptor activity by cholesterol (via direct binding to TM domains or cholesterol recognition amino acid consensus [CRAC] motifs) (*Karasawa et al., 2017*; *Murrell-Lagnado, 2017*; *Robinson et al., 2014*), palmitoylation (*Di Virgilio et al., 2018*; *Gonnord et al., 2009*; *Karasawa et al., 2017*), cooperative interactions between intracellular N- and C-termini (*Allsopp and Evans, 2015*), and calcium-dependent calmodulin binding (*Roger et al., 2008*). The latter, however, appeared to be specific for rat P2X7 and was not found in the human isoform (*Roger et al., 2010*). In monocyte-derived human macrophages, current facilitation as well as inflammasome activation, IL-1β release, blebbing, PS flip, and membrane permeabilization were inhibited by phospholipase A2 (PLA2) and Cl$^-$ channel antagonists (*Janks et al., 2019*), and it was suggested that facilitation represents a downstream effect of P2X7-mediated PLA2 and Cl$^-$ channel activation. Single channel recordings of HEK293 cell-expressed rat P2X7Rs recently revealed an increased open probability as a result of ATP-evoked current facilitation (*Dunning et al., 2021*). Here, we also observed a faster onset of current signal upon the second ATP application while changes in the amplitude were less obvious. Importantly, for most ANAP-containing P2X7R constructs studied here, fluorescence changes mirrored this behavior, strongly suggesting that it is a receptor-intrinsic property and does not involve currents from downstream-activated channels, such as Ca$^{2+}$-activated Cl$^-$ channels or pannexins (*Dunning et al., 2021*; *Ousingsawat et al., 2015*; *Pelegrin and Surprenant, 2006*; *Riedel et al., 2007*). Interestingly, the K127* head domain mutant showed faster fluorescence than current changes even upon the first ATP application. Thus, ANAP in this position reports a movement or interaction that precedes channel opening and is most likely related to ligand binding. A similar result was observed for the TMRM-labeled F11*/S124C double mutant, where the onset of TMRM signal upon the first ATP application was faster than the current and ANAP fluorescence change (but coincided upon the second application). In contrast to TMRM, ANAP in position 124 showed fluorescence signals that paralleled current responses, suggesting that different fluorophores can report different processes, possibly due to differences in size and/or sensitivity to the environment. Supporting the idea that these fast fluorescence changes are related to ligand binding, they were only observed in mutants containing fluorophores near the ATP binding site. Fast ligand-induced fluorescence changes already in the non-facilitated P2X7 state imply that ligand binding is unaltered between the first and second activation and consequently, changes in channel gating account for the observed current facilitation.

Fluorescence signals recorded from ANAP in positions near the channel gate (*Pippel et al., 2017*) could result from different simultaneously occurring effects during channel opening and evidence for both, a shift in ANAP emission toward longer wavelengths (position 340) and dequenching (positions

339 and 341) were observed. Interestingly, ANAP in position 340 revealed significant differences in the fluorescence amplitudes between the first and second ATP application. An intriguing explanation would be that it detects a slowly or non-reversible conformational change after the first ATP application, which could facilitate subsequent gating movements and thereby account for current facilitation. However, T340* was the only construct that did not show a faster current onset upon the second ATP application, possibly because ANAP substitution in this critical position already strongly facilitated gating, as indicated by the large 'leak' currents, likely reflecting partial constitutive ligand-independent opening.

Based on the above observations, we propose that the faster activation upon the second ATP application is an intrinsic property of the P2X7R. This conclusion is also in good agreement with the fact that the current facilitation but not downstream signaling events is seen in truncated P2X7 constructs (*Kopp et al., 2019*; *McCarthy et al., 2019*). One possibility for a molecular mechanism would be a pre-tensioning of TM2-helices during the first receptor activation that eases channel opening upon a second activation. It is not known, but likely that the cryo-EM structure of the ATP-bound open P2X7R represents the facilitated state. If so, the open-state stabilizing cap domain might not be locked in place in the naïve state but could be formed during the first receptor activation and then stabilized via the cysteine-rich anchor domain. The cap domain may then support the upward transition of TM2 and thereby accelerate current responses. Dynamic cysteine palmitoylation and cholesterol interactions might modulate this process as suggested before (*Di Virgilio et al., 2018*; *Dunning et al., 2021*; *Karasawa et al., 2017*; *Robinson et al., 2014*). Alternatively, initial receptor activation may change accessibility and/or affinity for a yet unknown allosteric ligand and thereby modulate P2X7 activation. All these suggested mechanisms are not mutually exclusive.

## Is the ballast domain affected by ATP-binding/channel opening?

While the functionality of P2X7 as a cation channel is not impaired by lack of the intracellular C-terminus (*Becker et al., 2008*; *Klapperstück et al., 2001*; *McCarthy et al., 2019*), its deletion disrupts a number of P2X7-mediated effects (*Kopp et al., 2019*), which most likely depend on downstream signaling pathways. A major aim of this study was the identification of C-terminal domains involved in such signaling. Most of the intracellular positions in which ANAP reported relative protein rearrangements were, however, located upstream of the cap domain either within the N-terminus (A3, C5, W7, F11) or right after TM2 (T361). Despite clear surface expression and current responses of at least 12 constructs with ANAP in the cytoplasmic ballast domain, only two of these mutants (D423* and A564*) revealed detectable but small fluorescence changes upon ligand application, suggesting that ATP binding induces only limited structural rearrangement in this domain, and that it is largely uncoupled from the extracellularly initiated conformational changes. Interestingly D423*, which showed only sporadic changes, lies in a short sequence with homology to an α-actinin 2 binding sequence (*Kim et al., 2001*). Since P2X7 activation induces plasma membrane morphology changes, and interactions with cytoskeletal proteins have been proposed (*Gu et al., 2009*; *Kim et al., 2001*; *Kopp et al., 2019*), an intriguing possibility would be that ANAP in position 423 reports interactions with cytoskeletal components. In A564*, ANAP is located near the GTP/GDP-binding site but showed much smaller signals than in positions near the ATP binding site, arguing against GTP/GDP (un-)binding, in agreement with the cryo-EM structures (*McCarthy et al., 2019*). However, A564 is also surrounded by other proposed interaction sites, including an LPS binding sequence and a calcium-dependent CaM binding motif (*Denlinger et al., 2001*; *Roger et al., 2010*; *Roger et al., 2008*), which might account for the observed signals.

In summary, we improved ANAP incorporation into *Xenopus* oocyte-expressed protein and performed an extensive VCF analysis of P2X7R mutants carrying ANAP in 61 positions throughout the receptor. We conclude from our data, that current facilitation is, at least partly, an intrinsic property of the P2X7R and involves an accelerated channel gating rather than ligand binding. In addition, we propose that ligand-induced extracellular and TM domain movements are not significantly translated to the cytosolic ballast domain and that intracellular ligands or interactors are required to 'activate' this domain. Protocols for simultaneous recording of ANAP with TMRM, $Ca^{2+}$-dependent R-GECO1.2, or mNeonGreen-labeled FRET partners are presented that might help to validate P2X7 downstream signaling events and analyze their molecular mechanisms and dynamics, once such interactors have been reliably determined.

## Materials and methods

### *Xenopus laevis* oocytes

*X. laevis* females were obtained from NASCO (Fort Atkinson, WI) and kept at the Core Facility Animal Models (CAM) of the Biomedical Center (BMC) of LMU Munich, Germany (Az:4.3.2–5682/LMU/BMC/CAM) in accordance with the EU Animal Welfare Act. To obtain oocytes, frogs were deeply anesthetized in MS222 and killed by decapitation. Surgically extracted ovary lobes were divided into smaller lobes and dissociated by ~2.5 hr incubation (16°C) with gentle shaking in ND96 solution (96 mM NaCl, 2 mM KCl, 1 mM CaCl$_2$, 1 mM MgCl$_2$, 5 mM HEPES, pH 7.4) containing 2 mg/ml collagenase (Nordmark, Uetersen, Germany) and subsequently defolliculated by washing (15 min) with Ca$^{2+}$-free oocyte Ringer solution (90 mM NaCl, 1 mM KCl, 2 mM MgCl$_2$, 5 mM HEPES). Stage V-VI oocytes were selected and kept in ND96 containing 5 µg/ml gentamicin until further use. In some cases, oocytes were commercially obtained (Ecocyte Bioscience, Dortmund, Germany), or ovaries were provided by Prof. Dr. Luis Pardo (Max Planck Institute for Experimental Medicine, Göttingen, Germany).

### cDNA and cloning

N-terminally His-tagged rat P2X1 cDNA in pNKS2 has been described (*Lörinczi et al., 2012*). An EGFP-tag was C-terminally added via a GSAGSA-linker sequence by Gibson assembly (*Gibson et al., 2009*) according to the protocol of the manufacturer (New England Biolabs GmbH, Frankfurt am Main, Germany).

cDNA encoding an N-terminally His-tagged rat P2X7R was subcloned into a pUC19 vector modified for cRNA expression in oocytes (termed pUC19o). pUC19o was generated by insertion (from 5` to 3`) of a synthesized T7 promoter sequence, a *Xenopus* globin 5'-UTR, and a Kozak sequence (*Kozak, 1987*) (GeneArt String DNA fragment, Life Technologies / Thermo Fisher Scientific Inc, Regensburg, Germany) and a 27 bp 3'-UTR (*Tanguay and Gallie, 1996*) followed by a poly A tail (51 adenines) obtained from the pNKS2 vector (*Gloor et al., 1995*) (for details of the UTRs see Key resource table).

The cDNA sequence of the aminoacyl-tRNA synthetase was obtained from the plasmid pANAP (Addgene #48696) (*Chatterjee et al., 2013*) and subcloned via Gibson assembly into pUC19o.

The coding sequence of *X. laevis* eRF1 (NCBI Reference Sequence: NM_001090894.1) with an E55D mutation (GeneArt String DNA fragment, Life Technologies/Thermo Fisher Scientific Inc, Regensburg, Germany) was cloned into pNKS2 via Gibson assembly. For recombinant expression in *E. coli*, the coding sequence of His-eRF1(E55D) was cloned into a modified pET28a vector via Gibson assembly.

Site-specific mutagenesis was performed with the Q5 Site-Directed Mutagenesis Kit (based on PCR-amplification) according to the manufacturer's protocol (New England Biolabs GmbH, Frankfurt am Main, Germany). Oligonucleotides were ordered from metabion GmbH (Planegg/Steinkirchen, Germany).

All constructs contained either an *ochre* (TAA) or *opal* (TGA) stop codon for normal translational termination to avoid C-terminal ANAP incorporation and read-through and were confirmed by sequencing (Eurofins Genomics, Ebersberg, Germany).

### eRF1 protein preparation

NiCo(DE3) bacteria were transformed with His-eRF1(E55D) in pET28a. 5 ml of a LB-Kanamicin pre-culture (~12 hr) was added to 300 ml ZY-5052 autoinduction media (*Studier, 2005*) supplemented with 100 µg/ml Kanamycin and grown for 6 hr at 37°C. The temperature was then reduced to 25°C, and bacteria were grown for another 18 hr. After pelleting by centrifugation (6500 g, 20 min) cells were resuspended in 40 ml lysis buffer (50 mM TRIS (tris(hydroxymethyl)aminomethane)-HCl, pH 8.0, 50 mM NaCl, 5 mM MgCl$_2$, 10% (v/v) glycerol, 0.1% (v/v) Triton X-100, 10 µg/ml DNase I, 100 µg/ml lysozyme), and sonicated (Bandelin Sono plus, TT13 cap, 50% duty cycle, 50% power) for 5 min in an ice bath. The lysate was pelleted at 40,000 × g (1 hr at 4°C). The supernatant was filtered (0.2 µm) and applied onto a Ni-NTA column (HisTrap FF, 5 ml, GE Healthcare Europe GmbH, Freiburg, Germany). Bound protein was washed with 10 column volumes of washing buffer (25 mM TRIS-HCl, pH 7.8, 500 mM NaCl, 20 mM imidazole, 0.25% [v/v] Tween 20, 10% [v/v] glycerol) and eluted with 6 column volumes of elution buffer (25 mM TRIS-HCl, pH 7.8, 500 mM NaCl, 300 mM imidazole, 0.25% [v/v] Tween 20 [v/v], 10% [v/v] glycerol). The eluate was concentrated (Amicon Ultra-15, 10 kDa MWCO, Millipore/Merck KgaA, Darmstadt, Germany), and buffer was exchanged by low-salt buffer (20 mM TRIS, 100 mM NaCl, pH 7.5) for subsequent anion exchange chromatography on a 5 ml Mono-Q

column (GE Healthcare Europe GmbH, Freiburg, Germany). Following an elution gradient with high-salt buffer (20 mM TRIS, 1 M NaCl, pH 7.5), protein-containing fractions were pooled, concentrated, and buffer was exchanged (1× PBS with 500 mM NaCl) for size exclusion chromatography on a Superdex 75 Increase (10/300). Purified His-eRF1(E55D) was shock-frozen in 10 µl aliquots and stored at –80°C.

## cRNA synthesis and tRNA

To prepare templates for cRNA synthesis, plasmids were linearized with EcoRI-HF (pNKS2) or NotI-HF (pUC19o) from New England Biolabs GmbH (Frankfurt am Main, Germany) and purified via MinElute Reaction Cleanup columns (Qiagen, Hilden, Germany) according to the manufacturer's protocol. Alternatively, templates (including the 5'-terminal RNA polymerase promoter site (T7 or SP6) and the 3'-terminal poly A) were amplified by PCR and purified using the NucleoSpin Gel and PCR Clean-up Kit (Macherey-Nagel, Düren, Germany) according to the manufacturer's protocol.

Capped cRNA was synthesized using the mMESSAGE mMACHINE SP6 or T7 Transcription Kits (Invitrogen/Thermo Fisher Scientific Inc, Schwerte, Germany), precipitated with LiCl, and dissolved in nuclease-free water (1 µg/µl if not stated otherwise).

The *amber* suppressor tRNA sequence was translated from the plasmid pANAP (Addgene #48696) (*Chatterjee et al., 2013*), provided with an universal 3'-terminal CCA-sequence (important for tRNA aminoacylation and translation), and chemically synthesized and purified via PAGE and HPLC (biomers. net GmbH, Ulm, Germany).

## Oocyte injection and ANAP incorporation

A Nanoject II injector (Science Products GmbH/Drummond, Hofheim, Germany) was used for nuclear and cytoplasmic injections.

cRNAs encoding cysteine-substituted receptors for TMRM labeling were injected as described (*Lörinczi et al., 2012*). Two different procedures were used for incorporation of ANAP:

The 2-step injection method was performed according to *Kalstrup and Blunck, 2017* using the plasmid pANAP that encodes the co-evolved, orthogonal, and ANAP-specific *amber* suppressor tRNA/tRNA synthetase pair (Addgene #48696 *Chatterjee et al., 2013*). 9.2 nl of pANAP (0.1 µg/µl) per oocyte were injected into the nucleus. 1–2 days later, 46 nl of an injection mix containing 0.20–0.25 µg/µl receptor-encoding cRNA (with or without an UAG codon at the site of interest) and 0.2–1.0 mM ANAP (L-ANAP trifluoroacetic salt or L-ANAP methyl ester, both AsisChem Inc, Waltham, MA) were injected into the cytoplasm.

The 1-step injection method was performed as described before (*Durner and Nicke, 2022*) with addition of mutated *X. laevis* eRF1 as indicated. An injection master mix comprising 0.25 mM ANAP TFA, 0.25 µg/µl cRNA encoding *X. laevis* eRF1 E55D, 0.2 µg/µl cRNA encoding the tRNA synthetase, and 0.4 µg/µl tRNA was freshly prepared. Three parts of the injection master mix were added to one part of 1 µg/µl receptor-encoding cRNA (with or without an UAG codon). 50.6 nl per oocyte were injected into the cytoplasm. Uninjected oocytes and oocytes injected with wt receptor cRNA served as negative and positive controls, respectively. Nuclease free water served as a substitute for individual components in control groups.

To optimize fUAA incorporation into *X. laevis* oocyte-expressed receptors, different procedures, concentrations of substances, and injection time points were compared (*Figure 1—figure supplement 1*). To optimize the concentrations of an individual component, the ratios and concentrations of the other components, as well as the expression times and receptor cRNA concentrations were kept constant in individual experiments. In cases where oocytes were incubated in membrane-permeable L-ANAP methyl ester, a 2 µM concentration in ND96 buffer (see below) was used.

Injected oocytes were kept in ND96 (96 mM NaCl, 2 mM KCl, 1 mM MgCl$_2$, 1 mM CaCl$_2$, 5 mM HEPES, pH 7.4–7.5) supplemented with gentamicin (50 µg/ml) at 16°C for at least 2 days.

## Receptor purification and SDS-PAGE

To evaluate plasma membrane expression of truncated and full-length His-tagged P2X7R mutants, surface-expressed receptors were fluorescently labeled, purified, and analyzed by SDS-PAGE. Three days after injection, 10 oocytes per group were labeled for 30–60 min (in the dark under rotation) in 200 µl 0.003% (m/V) aminoreactive, membrane-impermeant Cy5 Mono NHS Ester (Merck /

Sigma-Aldrich, Taufkirchen, Germany, diluted from a 1% [m/V] stock in DMSO) in ND96 (pH 8.5, 4°C) and then washed in ND96. Bright blue-stained damaged oocytes were then discarded, and intact oocytes were homogenized with a 200 µl pipet tip in 10 µl homogenization buffer per oocyte (0.1 M sodium phosphate buffer, pH 8.0, containing 0.4 mM Pefabloc SC and 0.5% $n$-dodecyl-β-D-maltoside, [both Merck/Sigma-Aldrich, Taufkirchen, Germany]). Membrane proteins were extracted by 10 min incubation on ice and separated from the debris by two centrifugation steps (10 min at 14,000 × g and 4°C). 100 µl of the protein extract were then supplemented with 400 µl of homogenization buffer containing 10 mM imidazole and added to 50 µl Ni$^{2+}$-NTA agarose beads (Qiagen GmbH, Hilden, Germany) preconditioned with washing buffer (0.1 M sodium phosphate buffer [pH 8.0] containing 0.08 mM Pefabloc, 0.1% $n$-dodecyl-β-D-maltoside, and 25 mM imidazole). After 1 hr incubation under inversion at 4°C in the dark, beads were washed three to four times with 500 µl washing buffer, and His-tagged protein was eluted (≥10 min at RT with occasional flipping to suspend the beads) with 2×50 µl elution buffer (20 mM Tris-HCl, 300 mM imidazole, 10 mM EDTA, and 0.5% $n$-dodecyl-β-D-maltoside). 32 µl of the eluate were supplemented with 8 µl 5× lithium dodecyl sulfate (LiDS) sample buffer (5% [w/v] LiDS, 0.1% bromphenol blue, 100 mM dithiothreitol, 40% [v/v] glycerol in 0.3 M Tris HCl [pH 6.8]), incubated at 95°C for 10 min, and separated by reducing SDS-PAGE on an 8% gel. Fluorescence-labeled protein was visualized with a Typhoon trio fluorescence scanner (GE Healthcare, Chicago, IL), and relative protein quantities were determined using FIJI (*Schindelin et al., 2012*). Lanes were selected as regions of interest and transformed into 1D profile plots. Band intensities were then quantified by integrating the area of each peak in the profile plot relative to the baseline of each lane. Data was visualized using GraphPad Prism software (Version 9.3.0, San Diego, CA).

## VCF recordings

Recordings were performed in a custom-made measuring chamber (*Figure 3*) that is split into an upper and lower compartment, which are individually perfused and connected by a 0.75 mm hole on which the oocyte is placed. The lower compartment has a transparent bottom, and the chamber was mounted on an Axiovert 200 inverted fluorescence microscope (Carl Zeiss Microscopy LLC, Oberkochen, Germany) so that the oocyte was centered above the objective with the animal pole facing down to avoid increased background fluorescence by the lighter vegetal pole. Upper and lower compartments were separately perfused with recording solution and recording or agonist solution, respectively, using a gravity-based perfusion system and a membrane vacuum pump. Solutions in the lower compartment were switched by computer-controlled magnetic valves.

To avoid inhibition by Ca$^{2+}$ or Mg$^{2+}$ and Ca$^{2+}$-mediated downstream effects and to obtain reproducible current responses, recordings were performed in divalent-free buffer (90 mM NaCl, 1 mM KCl, 5 mM HEPES, pH 7.4–7.5) complemented with flufenamic acid and ethylene glycol tetraacetic acid (EGTA) (both 0.1 mM). For measurements with Ca$^{2+}$-containing buffers, EGTA was omitted, and Ca$^{2+}$ (0.2–0.5 mM) was added (in case of P2X7-R-GECO constructs, FRET measurements between ANAP and mNeonGreen and control measurements of ANAP-containing constructs to test for Ca$^{2+}$-specific effects). If not otherwise noted, the agonist solution contained 300 µM ATP and was applied for 15 s in 195 s intervals. Intracellular electrode resistances were below 1.2 MΩ, and recordings were performed at room temperature at a holding potential of –30 mV to keep the current amplitudes reproducible. The solution exchange in the lower chamber is finished in about 1 s (*Lörinczi et al., 2012*).

To exclude mechanically induced fluorescence changes due to solution switching, all recording protocols started with sequential applications of ATP-free recording solutions from different tubes and magnetic valves. If required, solution speed and oocyte position were readjusted to ensure the absence of mechanical artifacts.

For fluorescence recordings, the microscope was equipped with two LEDs as excitation sources (UV-LED M365LP1 with 365 nm, green LED M565L3 with 565 nm, both Thorlabs GmbH, Bergkirchen, Germany). Since UV excitation in oocytes causes relatively high background fluorescence levels, detectors must feature a wide dynamic range, while maintaining a sufficiently high sensitivity in order to record small fluorescence changes. To this end, two cooled, high-sensitivity MPPC detectors (Hamamatsu Photonics K.K., Japan) were used for simultaneous fluorescence detection at two different spectral segments. For optical filters and dichroic mirrors see Key resource table.

Single-channel fully programmable instrumentation amplifiers with Bessel low-pass filter characteristics (Alligator Technologies, Costa Mesa, CA) were used for signal scaling. To minimize

photobleaching, LEDs were pulsed using self-developed high-speed LED drivers with sub-µs rise time. Pulse lengths were set in the ~20 µs range to allow for the fluorescence readout signal chain to settle. Fluorescence signal digitization was synchronized to the excitation pulses using an STM32F407 microcontroller (STMicroelectronics, Geneva, Switzerland). Its timer peripherals were re-triggered by each ADC conversion cycle in order to create an LED illumination pulse that starts shortly before the next ADC conversion cycle. Whenever two excitation wavelengths were used, excitation pulses were staggered in time with the longer wavelength excitation pulse signal being digitized first, preventing bleedthrough of background fluorescence excited by the shorter excitation wavelength to the longer-wavelength detection channel. A water-immersion objective with high numerical aperture and a large working distance (W N-Achroplan 63×/0,9 M27, Carl Zeiss Microscopy LLC, Oberkochen, Germany) was used to maximize the collection of emitted photons and to focus on the oocyte membrane.

Currents were measured with a Turbo Tec-05X amplifier and CellWorks E 5.5.1 software (both npi electronic GmbH, Tamm, Germany) and were used for current and fluorescence recordings and valve control. Current signals were digitized at 400 Hz and downsampled in CellWorks to 200 Hz.

## Dose-response analysis

To determine agonist dose-response curves, ATP was applied for 15 s in 195 s intervals. A reference concentration ($ATP_{Ref}$) of 300 µM was applied until stable responses were obtained and was then alternately applied with ATP concentrations ranging from 10 µM to 3 mM ($ATP_{Test}$). All responses were normalized to the response of $ATP_{Ref}$, and $EC_{50}$ values were calculated using the four-parameter Hill equation: *% Response = Bottom + (Top−Bottom)/(1+10^[(LogEC_{50}−X)\*n_H])* with *Bottom* and *Top* constrained to 0%, *and* maximum responses, respectively, *X* corresponding to the log of agonist concentration, and $n_H$ corresponding to the Hill coefficient.

## Data analysis

Fluorescence and current signals were analyzed and visualized using a Python-based script (for packages used, see Key resource table): Fluorescence signals were denoised using a fifth-order Bessel filter with a low-pass corner frequency of 4 Hz. Maximum amplitudes of ATP-evoked current and fluorescence responses from different receptor constructs were summarized, compared, and visualized using GraphPad Prism software (Version 9.3.0, San Diego, CA). The following inclusion criteria were applied for recordings:

(i) ATP application must evoke a current response >0.1 µA, (ii) leak currents must be stable for the duration of the recording (at least two ATP applications), (iii) repeated ATP applications must elicit reproducible current responses (>0.8 µA), (iv) fluorescence signals must be without mechanical artifacts and clearly distinguishable from fluorescence changes of wt expressing oocytes (see below). 2–3 days after injection, repeated application of 300 µM ATP to wt-expressing oocytes elicited reproducible currents (i.e. first and second current responses differed less than 10% and reached a plateau, at least during the second application), which were taken as a reference. Longer expression times resulted in irregular and irreproducible current responses and less stable oocytes. In case of mutated receptors, longer expression times were often needed to yield current responses comparable to wt P2X7.

We observed a gradual decrease in fluorescence signal for the duration of ligand application in control oocytes expressing wt receptors even in the absence of ANAP. To distinguish ANAP-specific fluorescence signals from these gradual changes, for signal analysis only fluorescence changes upon ATP application were considered that were either positive, or negative but additionally not linear. If fluorescence signals from mutant expressing oocytes were not distinguishable from fluorescence changes observed for wt expressing oocytes no fluorescence change was assumed (0% ΔF/F). Only signals that were recorded in at least three different oocytes were considered for analysis. Additionally, fluorescence changes that were recorded in less than 40% of analyzed oocytes expressing one specific receptor construct or that had averaged ΔF/F values <0.3% were not considered.

## Statistical analysis

Data were either represented as mean ± S.D., as box plots, or as mean ± S.E.M. with the number of recordings given in brackets, and statistical analysis was performed by either two-tailed unpaired Welch's *t*-test or two-tailed paired Student's *t*-test, as indicated. Values of p<0.05 were defined as

statistically significant with *, **, ***, and **** denoting values of $p<0.05$, 0.005, 0.0005, and 0.0001 or 0.00005, respectively.

## Data availability

All data generated or analyzed during this study are included in the manuscript and supporting files. Original VCF recordings, extracted VCF data, and scans from SDS-PAGE gels are provided as source data files for *Figure 1*, *Figure 1—figure supplement 1*, *Figure 2*, *Figure 2—figure supplement 1*, *Figure 3*, *Figure 3—figure supplement 1*, *Figure 3—figure supplement 2*, *Figure 3—figure supplement 3*, *Figure 4*, *Figure 4—figure supplement 1*, *Figure 5*, *Figure 5—figure supplement 1*, *Figure 5—figure supplement 2*, *Figure 5—figure supplement 3*, *Figure 6*, *Figure 6—figure supplement 1*, *Figure 6—figure supplement 2*, *Figure 6—figure supplement 3*, *Table 1*, and *Table 2*. The source data files of *Table 1* include source data of *Figure 2—figure supplement 1*, *Figure 3*, *Figure 4*, and *Figure 5* and are assigned accordingly in *Table 1—source data 1*. The original recordings of *Table 1* have been deposited with Dryad (DOI https://doi.org/10.5061/dryad.p8cz8w9tb). The source data files of *Table 2* include source data of *Figure 3—figure supplement 3* and *Figure 5—figure supplement 2*.

Note that original current and fluorescence recordings provided as comma separated value files each contain three columns of values (from left to right): (1) current values, (2) fluorescence signals of longer emission wavelengths, and (3) fluorescence signals of shorter emission wavelengths.

## Acknowledgements

This work was supported by the Deutsche Forschungsgemeinschaft (DFG, German Research Foundation, Project-ID: 335447717 - SFB 1328, A15). We thank Luis Pardo, Kerstin Dümke, and Monika Haberland for providing *Xenopus laevis* oocytes.

## Additional information

### Funding

| Funder | Grant reference number | Author |
|---|---|---|
| Deutsche Forschungsgemeinschaft | 335447717 - SFB 1328, Project A15 | Annette Nicke |

The funders had no role in study design, data collection and interpretation, or the decision to submit the work for publication.

### Author contributions

Anna Durner, Conceptualization, Formal analysis, Investigation, Methodology, Writing - original draft, Writing - review and editing; Ellis Durner, Conceptualization, Resources, Formal analysis, Methodology, Writing - original draft, Writing - review and editing; Annette Nicke, Conceptualization, Supervision, Funding acquisition, Writing - original draft, Project administration, Writing - review and editing

### Author ORCIDs

Anna Durner ⓘ http://orcid.org/0000-0002-0993-8869
Ellis Durner ⓘ http://orcid.org/0000-0002-4461-9257
Annette Nicke ⓘ http://orcid.org/0000-0001-6798-505X

### Decision letter and Author response

Decision letter https://doi.org/10.7554/eLife.82479.sa1
Author response https://doi.org/10.7554/eLife.82479.sa2

## Additional files

### Supplementary files

• MDAR checklist

## Data availability

All data generated or analyzed during this study are included in the manuscript and supporting files. Original recordings or scans from SDS-PAGE gels are provided as source data with the respective figures. Original recordings of Table 1 are deposited with Dryad (for data assignment see Table 1-source data 1). This paper does not report original code.

The following dataset was generated:

| Author(s) | Year | Dataset title | Dataset URL | Database and Identifier |
|---|---|---|---|---|
| Nicke A, Durner A, Durner E | 2023 | Table 1-source data 1 (original recordings) | http://dx.doi.org/10.5061/dryad.p8cz8w9tb | Dryad Digital Repository, 10.5061/dryad.p8cz8w9tb |

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

# Appendix 1

## Appendix 1—key resources table

| Reagent type (species) or resource | Designation | Source or reference | Identifiers | Additional information |
|---|---|---|---|---|
| Recombinant DNA reagent | cDNA *Xenopus laevis* eRF1(E55D) | NCBI Reference Sequence: NM_001090894.1, Life Technologies / Thermo Fisher Scientific | | GeneArt String DNA fragment (cloned into pET28a and pNKS2) |
| Recombinant DNA reagent | cDNA *Rattus norvegicus* P2X7 | | | in modified pUC19 (pUC19o) |
| Recombinant DNA reagent | cDNA *Rattus norvegicus* P2X1 | *Lörinczi et al., 2012* | | in pNKS2 |
| Recombinant DNA reagent | cDNA *Rattus norvegicus* calmodulin-1 | UniProt: PDP29; NCBI Reference Sequence: NM_031969.3 | | Codon-optimized for *Xenopus laevis* (Invitrogen / Thermo Fisher Scientific), C-terminally linked to *Rattus norvegicus* P2X7 via GS-linker (ggatct) |
| Recombinant DNA reagent | Plasmid: pNKS2 | *Gloor et al., 1995* | | |
| Recombinant DNA reagent | Plasmid: pUC19 | New England Biolabs GmbH | CAT# N3041S | |
| Recombinant DNA reagent | Plasmid: pANAP | *Chatterjee et al., 2013* | CAT#48696 | |
| Recombinant DNA reagent | EGFP | *Yang et al., 1996* | | the original enhanced GFP, mammalian codon-optimized, C-terminally linked to *Rattus norvegicus* P2X1 via GSAGSA-linker sequence (ggatctgcaggatctgca) |
| Recombinant DNA reagent | R-GECO1.2 | *Wu et al., 2013* | Addgene #45494 | |
| Recombinant DNA reagent | mNeonGreen | *Shaner et al., 2013* | | Codon-optimized for *Xenopus laevis* (Invitrogen / Thermo Fisher Scientific) |
| Recombinant DNA reagent | M13-like peptide from CaM-dependent kinase | *Rattus norvegicus* myosin light chain kinase, smooth muscle; Uniprot: D3ZFU9 | | RRKWQKTGNAVRAIGRLSSM; cloned between *Rattus norvegicus* calmodulin-1 and mNeonGreen with N- and C-terminal GS-linkers (ggcagc and ggatct, respectively) |
| Sequence-based reagent | Oligonucleotides | metabion GmbH | | |
| Sequence-based reagent | *Amber* suppressor tRNA, synthesized oligonucleotide, sequence derived from pANAP, an universal 3'-terminal CCA-sequence was added: 5'-gcc cgg aug gug gaa ucg gua gac aca agg gau ucu aaa ucc cuc ggc guu cgc gcu gug cgg guu caa guc ccg cuc cgg gua cca –3' | biomers.net GmbH; *Chatterjee et al., 2013*; *Durner and Nicke, 2022* | | |
| Sequence-based reagent | Sanger sequencing | Eurofins Genomics, https://eurofinsgenomics.eu/ | | |
| Sequence-based reagent | 5'-UTR, GeneArt String DNA fragment (cloned into pUC19 (small letters) before the start codon (italic letters)), gtacccggggatcctctTAATACGACTCACTATAGGCTTGT TCTTTTTGCAGA AGCTCAGAATAAACGCTCAACTTTGGCTCGAG GCCACC*atg* | Life Technologies / Thermo Fisher Scientific, *Kozak, 1987* | | |
| Sequence-based reagent | 3'-UTR, (cloned into pUC19 (small letters) after the stop codon (italic letters)), *tga*CCCAAAACAAAAACGGAATATG CAAACAAAAAAAAAAAAAAAAAAAAAAA AAAAAAAAAAAAAAAAAAAAAAAAAAAA GAATTCTAGAGCGGCCGCagagtcgacctgcagg | pNKS2, *Gloor et al., 1995* | | |

*Appendix 1 Continued on next page*

*Appendix 1 Continued*

| Reagent type (species) or resource | Designation | Source or reference | Identifiers | Additional information |
|---|---|---|---|---|
| Peptide, recombinant protein | EcoRI-HF | New England Biolabs GmbH | CAT#R3101S | |
| Peptide, recombinant protein | NotI-HF | New England Biolabs GmbH | CAT#R3189S | |
| Commercial assay or kit | Gibson Assembly Master Mix | New England Biolabs GmbH | CAT#E2611L | |
| Commercial assay or kit | Q5 Site-Directed Mutagenesis Kit | New England Biolabs GmbH | CAT#E0552S | |
| Commercial assay or kit | MinElute Reaction Cleanup Kit | QIAGEN GmbH | CAT#28204 | |
| Commercial assay or kit | | | | |
| Commercial assay or kit | Macherey-Nagel NucleoSpin Gel and PCR Clean-up Kit | Fisher Scientific / Thermo Fisher Scientific | CAT# 11992242 | |
| Commercial assay or kit | mMESSAGE mMACHINE T7 Transcription Kit | Invitrogen / Thermo Fisher Scientific | CAT# AM1344 | |
| Commercial assay or kit | mMESSAGE mMACHINE SP6 Transkription Kit | Invitrogen / Thermo Fisher Scientific | CAT#AM1340 | |
| Chemical compound, drug | ATP disodium salt hydrate | Sigma-Aldrich | Cat#A3377 | |
| Chemical compound, drug | L-ANAP trifluoroacetic salt | AsisChem Inc. | Cat#ASIS-0014 | |
| Chemical compound, drug | L-ANAP methyl ester | AsisChem Inc. | Cat#ASIS-0146 | |
| Chemical compound, drug | Collagenase NB 4 G proved grade | Nordmark Pharma GmbH | Cat#S1746502 | |
| Chemical compound, drug | Gentamicin sulfate | Roth | CAT#0233.4 | |
| Chemical compound, drug | Cy5 Mono NHS Ester | Merck / Sigma-Aldrich | CAT#GEPA15101 | |
| Chemical compound, drug | Pefabloc SC | Merck / Sigma Aldrich | CAT#76307 | |
| Chemical compound, drug | n-Dodecyl-β-D-Maltoside, ULTROL grade | Merck / Sigma Aldrich | CAT#324355 | |
| Chemical compound, drug | Ni-NTA Agarose | QIAGEN GmbH | CAT#1018244 | |
| Chemical compound, drug | Flufenamic acid | Merck / Sigma Aldrich | CAT#F9005 | |
| Chemical compound, drug | 0.5 M EDTA ph 8.0 | Thermo Scientific | CAT#R1021 | |
| Chemical compound, drug | TMRM | Biomol | CAT#ABD-419 | |
| Chemical compound, drug | A 438079 hydrochloride | TOCRIS | CAT#2972 | |

*Appendix 1 Continued on next page*

*Appendix 1 Continued*

| Reagent type (species) or resource | Designation | Source or reference | Identifiers | Additional information |
|---|---|---|---|---|
| Software, algorithm | CellWorks E 5.5.1 | npi electronic, http://cellworks.de/ | | |
| Software, algorithm | PyMOL | http://www.pymol.org/ | RRID:SCR_000305 | |
| Software, algorithm | Python Programming Language 3.10.4 | http://www.python.org/ | RRID:SCR_008394 | |
| Software, algorithm | NumPy 1.22.3 | http://www.numpy.org | RRID:SCR_008633 | |
| Software, algorithm | MatPlotLib 3.5.1 | http://matplotlib.sourceforge.net | RRID:SCR_008624 | |
| Software, algorithm | SciPy 1.8.0 | http://www.scipy.org/ | RRID:SCR_008058 | |
| Software, algorithm | GraphPad Prism 9.3.0 and 9.5.0 | http://www.graphpad.com/ | RRID:SCR_002798 | |
| Software, algorithm | (Fiji Is Just) ImageJ 2.3.0 | *Schindelin et al., 2012*, http://fiji.sc | RRID:SCR_002285 | |
| Other | Turbo Tec-05X Amplifier | npi electronic GmbH | CAT#TEC-05X | VCF-Setup components, electronics |
| Other | PCI-6221, DAQ, Multifunction I/O Device, 16-Bit | National Instruments | CAT# 779066–01 | VCF-Setup components, electronics |
| Other | Single-channel fully programmable Instrumentation Amplifier Low Pass Filter, USBPGF-S1/L with 8th pole Bessel filter characteristics | Alligator Technologies | CAT#USBPGF-S1/L | VCF-Setup components, electronics |
| Other | 2 x MPPC modules | Hamamatsu Photonics K.K. | CAT#C13366-3050GA | VCF-Setup components, electronics |
| Other | Power adapter/linear regulator | KNIEL System-Electronic GmbH | Custom-made | VCF-Setup components, electronics |
| Other | Axiovert 200 inverted fluorescence microscope | Carl Zeiss Microscopy LLC | | VCF-Setup components, optics |
| Other | Objektiv W N-Achroplan 63 x/0,9 M27 | Carl Zeiss Microscopy LLC | CAT#420987-9900-000 | VCF-Setup components, optics |
| Other | M565L3, mounted LED at 565 nm | Thorlabs GmbH | CAT#M565L3 | VCF-Setup components, optics |
| Other | M365LP1, Mid Power Mounted LED at 365 nm | Thorlabs GmbH | CAT#M365LP1 | VCF-Setup components, optics |
| Other | 2× lenses for LED collimation | Thorlabs GmbH | CAT#ACL2520U-A | VCF-Setup components, optics |
| Other | ET555/20×, 25 mm Dia Mounted, Single Bandpass Filter (for excitation) | Chroma Technology GmbH | CAT# IN026697 | VCF-Setup components, optics |
| Other | ET365/20×, 25 mm Dia Mounted, Single Bandpass Filter (for excitation) | Chroma Technology GmbH | CAT# IN053211 | VCF-Setup components, optics |
| Other | T387lp, 25.5×36×1 mm, Longpass Dichroic Beamsplitter | Chroma Technology GmbH | CAT# IN040921 | VCF-Setup components, optics |
| Other | 79003bs, Multi Dichroic Beamsplitter | Chroma Technology GmbH | CAT# CS294227 | VCF-Setup components, optics |
| Other | 59002bs, Multi Dichroic Beamsplitter | Chroma Technology GmbH | CAT# IN040206 | VCF-Setup components, optics |
| Other | T425lpxr,25.5×36×1 mm, Longpass Dichroic Beamsplitter | Chroma Technology GmbH | CAT# IN025246 | VCF-Setup components, optics |
| Other | Relay lense | Thorlabs GmbH | CAT#AC254-060-A | VCF-Setup components, optics |
| Other | DMLP550R, Longpass Dichroic Beamsplitter | Thorlabs GmbH | CAT#DMLP550R | VCF-Setup components, optics |
| Other | T470lpxr, Longpass Dichroic Beamsplitter | Chroma Technology GmbH | CAT# IN030502 | VCF-Setup components, optics |

*Appendix 1 Continued on next page*

*Appendix 1 Continued*

| Reagent type (species) or resource | Designation | Source or reference | Identifiers | Additional information |
|---|---|---|---|---|
| Other | T495lpxr, 25.5×36×1 mm, Longpass Dichroic Beamsplitter | Chroma Technology GmbH | CAT# IN005752 | VCF-Setup components, optics |
| Other | ET490/40×, 25 mm Dia Mounted (for emission) | Chroma Technology GmbH | CAT# IN039532 | VCF-Setup components, optics |
| Other | ET610/75 m, 25 mm Dia Mounted (for emission) | Chroma Technology GmbH | CAT# IN036520 | VCF-Setup components, optics |
| Other | ET455/50 m (for emission) | Chroma Technology GmbH | CAT# IN067607 | VCF-Setup components, optics |
| Other | ET500lp, 25 mm Dia Mounted (for emission) | Chroma Technology GmbH | CAT# IN006640 | VCF-Setup components, optics |
| Other | MF460-60 (for emission) | Thorlabs GmbH | CAT#MF460-60 | VCF-Setup components, optics |
| Other | 2× lenses for focusing on MPPC | Thorlabs GmbH | CAT#LB1761-A | VCF-Setup components, optics |

