## [Editor Report]

This manuscript constitutes a valuable foray into the conformational rearrangements throughout various domains of the notoriously difficult-to-study P2X7 receptor, with a focus on the enigmatic intracellular 'ballast' domain. The molecular origin of the facilitation process and effects by intracellular factors will require future study, but the authors provide convincing evidence that the ballast domain is unlikely to undergo major conformational changes upon ATP-induced gating. The work is of interest to those interested in the role of enzymatically active intracellular domains of membrane proteins.

---

## [Decision Letter]

**Decision letter after peer review:**

Thank you for submitting your article "Voltage clamp fluorometry analysis of the P2X7 receptor suggests a limited conformational interplay between extracellular ATP binding and the intracellular ballast domain" for consideration by *eLife*. Your article has been reviewed by 3 peer reviewers, and the evaluation has been overseen by a Reviewing Editor and Richard Aldrich as the Senior Editor. The following individual involved in review of your submission has agreed to reveal their identity: Gucan Dai (Reviewer #3).

Essential revisions:

The reviewers appreciated the effort needed to generate this impressive data set. Although there was enthusiasm for the overall approach and the reviewers agreed on the potential of the study, it was felt that additional mechanistic insight on facilitation and impact of intracellular factors would be required before the work would be of sufficient interest. Therefore we ask you to carefully address the following essential revisions:

(1) Examination of fluorescence changes in the ballast domain (e.g. A564*) in response to manipulation of at least one or two relevant cytoplasmic factors (e.g. GTP/GDP, Zn2+, palmitoylation (e.g. mutations) or Ca^2+^/CaM). This is crucial to provide mechanistic insight on the functional role of the ballast domain.

(2) Careful validation that the fluorescence changes from e.g. S124*, F11*, S127C do in fact track facilitation. This could be achieved by studying fluorescence signals of mutants that alter or eliminate facilitation (if those mutants exist) or by examining the effect of experimental manipulations of cytoplasmic factors on fluorescence (and current) facilitation (e.g. removal of extracellular Ca^2+^ or similar). This is particularly interesting for the intracellular N-terminus, which cannot be labeled with cysteine-reactive fluorophores.

(3) A more thorough characterization of key mutants in terms of ATP sensitivity of current and fluorescence, facilitation and activation/deactivation kinetics. This is required for a more quantitive analysis and has the potential to yield additional insight on the mechanism of receptor activation.

*Reviewer #1 (Recommendations for the authors):*

I would have expected the authors to functionally characterize the mutants on which they focus in the paper, for instance by measuring the ATP EC50, to have an idea of whether ANAP incorporation at the different sites perturbs receptor function.

When studying fluorescence changes at the level of the head domain (p7), the authors write that "fluorescence and current changes started simultaneously and showed shorter rising times upon repeated ATP applications"; "K127* showed a fluorescence change that was faster than the current increase" but no quantification of the fluorescence and current kinetics is given. It would be nice to have such quantification. It would be also nice to have a comparison of the fluorescence and current ATP dose-response curves for mutants showing sufficiently large fluorescence changes. This would further support the fact that ANAP at the level of the head domain tracks gating.

ANAP incorporation at TM2 positions seems to induce constitutive currents in P2X7Rs. Is there a channel blocker specific for P2X7Rs that would allow to give an estimation of constitutive activity of these mutants?

Page 9, it is written that "the kinetics of F11* fluorescence correlated with current facilitation". As before, this statement solely relies on the qualitative observation of current and fluorescent traces. Quantitative measurements of fluorescence and current kinetics during the first and second application of ATP is necessary to substantiate that claim. Here again, a dose-response curve of current and fluorescence for ATP of P2X7-F11* would be a plus to determine the gating step(s) tracked by ANAP at this position.

Attributing the lack of ATP-induced fluorescence changes observed in mutants with ANAP incorporated in the ballast domain to a lack of functional coupling between the extracellular and the intracellular domains is a strong assumption. The issue with ANAP incorporation in the C-terminus is the membrane expression of truncated proteins that are functional. Since the authors showed that these truncated proteins exists, if they are functional, they contribute to the measured current. The proportion of ANAP-labeled proteins could thus be too low to yield fluorescence changes despite large currents. To study the contribution of truncated proteins, the authors should characterize these truncated proteins by expressing just the mRNA with the amber stop codon in *Xenopus oocytes*, investigate potential ATP-induced currents carried by these truncated receptors and measure ATP dose-response curves. Comparing ATP dose-response curves of ANAP-mutants to purely truncated proteins should allow estimating the contribution of truncated proteins to the general current.

It seems that calcium induces fluctuations of ANAP fluorescence. Does calcium have any direct effect on ANAP fluorescence?

It seems that fluorescence changes at some positions of the extracellular head domain and intracellular N-terminus track the process of facilitation. Tracking of this process is probably the most interesting finding of this paper, since the mechanism of facilitation is not known. In the paper, facilitation is measured as the increase of channel opening kinetics upon multiple applications of ATP. However, the first application of ATP clearly shows a small, fast component and a bigger, slow component (see, for instance, Figure 3F, 5C, 6A). Could it be that the first component reflects initial gating and the second one the slow facilitation process that occurs upon prolonged ATP application? Could the fast kinetics of K217* fluorescence match the fast component of current, indicating that K217* tracks the initial ATP opening step but not current facilitation? Similarly, on Figure 6 it looks like TMRM kinetics match first the fast current component, while F11* fluorescence matches the kinetics of current facilitation. Could the authors comment on it?

*Reviewer #2 (Recommendations for the authors):*

To strengthen the points of weakness above and to elevate the scientific merits of the paper, I have some comments which require attention.

(1A) I am curious to know the P2X7 structural differences by VCF between just before the 1st application and just before the 2nd application of ATP, to clarify the structural background of "memory" of ATP application.

(1B) Will there be any difference in (1A), in the presence and absence of extracellular Ca^2+^, i.e. Ca^2+^ influx via P2X7.

(2A) Facilitation of F change was detected e.g. in S124*, F11*, S127C. Possible mechanisms of facilitation are discussed including mediators such as GDP, cholesterol, palmitoylation, Ca^2+^/CaM (Lines 335-342), However, there are no experimental results to support or deny them. It is desired to examine the effect of experimental manipulations of these factors on the F(and I) facilitation (e.g. removal of extracellular Ca^2+^).

(2B) Line 273: "the so-called current facilitation in P2X7 is due to a change in receptor gating rather than ligand binding", Line 366: "Ligand binding is unaltered between the 1st and 2nd activation and consequently, changes in channel gating account for the observed current facilitation". I wonder if this is correct, as F facilitation is observed in S124* which is close to the ATP binding site (Figure 3F).

(2C) F facilitation was detected not in S124C-TMRM (Figure 6A), not in K127* (Figure 3F), but in S124* (Figure 3F). I cannot fully understand the ground of differences.

(3) As to the function of the ballast domain, it is described "intracellular ligand or interactors are required to "activate" this domain" (line 421). A possible involvement of the ballast domain to the functional modulation of P2X7 by the binding of various factors such as GTP/GDP, LPS and Ca^2+^/CaM are discussed (lines 407-415). The authors showed limited structural rearrangements of the ballast domain upon ATP application, but there is no evidence by VCF to prove activation by intracellular ligand. I would like to see data of F change of such as A564* in the ballast domain in response to manipulation of intracellular factors.

(4) Figure 4C: Does the F facilitation in T340* truly reflect "facilitation", in spite of the absence of current facilitation (as described in line 376)? The authors explained that a lack of faster current in the 2nd application is due to "strongly facilitated gating" (line 377). If that is the case, why there is F facilitation?

(5) Do the F change of R-Geco (Figure 6B) and FRET change by mNeonGreen (Figure 6C) disappear in the absence of extracellular Ca^2+^?

*Reviewer #3 (Recommendations for the authors):*

1. Please cite this paper when talking about eRF-E55D as well, https://doi.org/10.7554/*eLife*.37248.

2. The 3'-terminal CCA-sequence is already in the pANAP plasmid. More elaboration on how it augmented the translation is needed to make it clear to readers.

3. Additional FRET experiments might be out of the scope of this paper, but please add more discussion and references regarding this possibility in the paper.

4. One area that could be improved on is the uncoupled "ballast" domain. How might GTP/GDP and dinuclear Zn2+-binding rearrange this domain? Can pharmacological manipulations be used to change intracellular GTP/GDP or Zn2+ levels to see how Anap labeled in this domain changes its fluorescence differently? Could the palmitoylation of the anchor domain play a role for uncoupling the "ballast' domain and for the current facilitation?

5. Line 634, should be "artifacts"?

---

## [Author Response]

Essential revisions:The reviewers appreciated the effort needed to generate this impressive data set. Although there was enthusiasm for the overall approach and the reviewers agreed on the potential of the study, it was felt that additional mechanistic insight on facilitation and impact of intracellular factors would be required before the work would be of sufficient interest. Therefore we ask you to carefully address the following essential revisions:(1) Examination of fluorescence changes in the ballast domain (e.g. A564*) in response to manipulation of at least one or two relevant cytoplasmic factors (e.g. GTP/GDP, Zn2+, palmitoylation (e.g. mutations) or Ca^2+^/CaM). This is crucial to provide mechanistic insight on the functional role of the ballast domain.

We agree that more insight in the role of the ballast domain would be highly informative. However, the relevance of the cytoplasmic factors such as Zn^2+^ and GTP has not been defined so far and no functional readouts for mutations in the GTP or Zn^2+^ binding sites are known, which hampers their investigation. At least in the Zn^2+^ binding site, we expected mutations to be not tolerated (Gonnord et al., 2009), likely because of a structural function in receptor folding. Also, our set-up and experimental conditions do not allow a direct comparison of effects before and after intracellular modulation (e.g. by Zn^2+^ or GDP/GTP).

We therefore concentrated on possible effects of (de-)palmitoylation and a previously described interaction with calmodulin and generated and analyzed the following three additional mutants:

(i) Cys-Ala P2X7 (replacement of six palmitoylated residues Ser360 and Cys362, 371, 373, 374, 377, McCarthy et al., 2019) in the cysteine-rich region by alanine residues,

(ii) DCys P2X7 (deletion of the cysteine-rich intracellular region (S360-C377)), and

(iii) DCaM P2X7 (disruption of a proposed calmodulin (CaM) binding site by three point mutations I541T, S552C, V559G Roger et al., 2010).

According to (McCarthy et al., 2019) and (Roger et al., 2010), DCys- and Cys-Ala mutants were expected to be not palmitoylated and to show no facilitation. In agreement with this hypothesis, we indeed did not see facilitation in the DCys mutant while expression of the Cys-Ala mutant was too low for analysis (Figure 3—figure supplement 3, compare also point 2). Expression of these mutants, was further reduced by the additional mutations for ANAP incorporation (F11*, S124*, K127*, D423*, or A564*), making a reliable detection of the corresponding fluorescence changes impossible.

The CaM-binding site in rat P2X7 was previously shown to affect current facilitation (Roger et al., 2010). Here, the DCaM mutant showed facilitation comparable to wt P2X7. Likewise, combination with ANAP in intracellular positions or the head domain (F11*, D423*, A564* and S124*, K127*, respectively) revealed similar fluorescence changes as observed for the single mutants with intact CaM binding sites. This argues against a major functional effect of the CaM binding site mutation on current kinetics or molecular movements in the oocyte-expressed receptor. These data are now included in Figure 5—figure supplement 3 and mentioned in line 286ff.

As an additional approach to address a potential interaction of the P2X7 receptor with CaM, we performed FRET experiments with mNeonGreen-tagged CaM and ANAP-labeled P2X7 (ANAP introduced in A3, Q455, L523, or L527 and preliminary experiments with R125, T340, R364, E406, V517, L569, and Y595). However, these recordings revealed no differences to the negative control (wt P2X7 with co-injected ANAP and mNeonGreen-tagged CaM) and were therefore discontinued.

These additional data are now added as Figure 6—figure supplement 3 and referred to in the text (lines 343ff).

Although these experiments cannot exclude an interaction with CaM, they do not support an important role of CaM in the observed current facilitation.

(2) Careful validation that the fluorescence changes from e.g. S124*, F11*, S127C do in fact track facilitation. This could be achieved by studying fluorescence signals of mutants that alter or eliminate facilitation (if those mutants exist) or by examining the effect of experimental manipulations of cytoplasmic factors on fluorescence (and current) facilitation (e.g. removal of extracellular Ca^2+^ or similar). This is particularly interesting for the intracellular N-terminus, which cannot be labeled with cysteine-reactive fluorophores.

Deletion of the cysteine-rich region was previously shown to eliminate current facilitation (Roger et al., 2010). In agreement, current rise times (compare also point 3 below) of the first and second ATP applications to the DCys mutant were unaltered in our experiments. In contrast, current rise times clearly accelerated in the wt, S124*, and F11* mutations, demonstrating that we can indeed identify current facilitation in our set-up. However, the DCys and Cys-Ala mutants showed already decreased currents and their combination with sites for ANAP incorporation (F11*, S124*) resulted in further reduced functional expression, which prevented analysis of the corresponding fluorescence changes.

An S127C mutant was not prepared or mentioned in the text and we assume that the reviewer referred to K127*, which showed facilitation only for the current but not for the fluorescence. For DCys/K127*, robust fluorescence changes were only detected in one out of 12 oocytes. In this case, the fluorescence changes were virtually identical upon first and second ATP application, as expected.

As an alternative approach, we also combined the three ANAP-substituted mutants (S124*, K127*, F11*) with an S23N mutant that has been shown to alter the kinetics of receptor activation in human P2X7 (Allsopp and Evans, 2015). However, this mutation did not induce an effect on rat P2X7 kinetics.

The analyses confirming facilitation of current rise times in the S124*, F11* mutations and elimination of current facilitation by the DCys mutation are now included in Figure 3—figure supplement 3 and in lines 190ff in the text.

Extracellular Ca^2+^ was generally omitted in the presented VCF recordings since it interfered with ANAP fluorescence measurements (compare line 324 and material and methods, line 653ff). Ca^2+^-containing buffer affected overall baseline fluorescence in VCF-experiments independent of the presence or absence of ANAP and additionally promoted receptor desensitization (Figure 6—figure supplement 1), making these conditions unsuitable for reliable detection and analysis of receptor responses.

We now added photometric data, demonstrating also a direct effect of Ca^2+^ on ANAP fluorescence (Figure 6—figure supplement 1), albeit only at very high concentrations.

(3) A more thorough characterization of key mutants in terms of ATP sensitivity of current and fluorescence, facilitation and activation/deactivation kinetics. This is required for a more quantitive analysis and has the potential to yield additional insight on the mechanism of receptor activation.

We generated ATP dose response curves for the current responses of wt P2X7 receptor, the key mutants F11*, S124*, K127*, D423*, A564*, and the TMRM-labeled F11*/S124C double mutant and show that their ATP sensitivities differed less than 3fold from that of the wt P2X7.

Unfortunately, dose response analysis of fluorescence responses could not be performed due to ANAP bleaching during the long recording protocol (about 15-40 min) and the high variability of background fluorescence, which made normalization of responses from different oocytes impossible.

For a quantitative analysis of current facilitation/activation kinetics, we determined the 10-50% rise times of current and fluorescent signal increases upon first and second ATP application for wt P2X7, the DCys mutant, the DCaM mutant, as well as mutants F11*, S124*, K127*, T340*, and the TMRM-labeled F11*/S124C double mutant. Kinetics of channel deactivation were not analyzed as they were strongly delayed, most likely due to turbulences in the measuring chamber that resulted in a slow wash-out of ATP.

These data are now provided in Figure 3—figure supplement 3, Figure 4—figure supplement 1, Figure 5—figure supplement 2 and 3, and Table 2. For corresponding text see specific reviewer questions below.

Reviewer #1 (Recommendations for the authors):I would have expected the authors to functionally characterize the mutants on which they focus in the paper, for instance by measuring the ATP EC50, to have an idea of whether ANAP incorporation at the different sites perturbs receptor function.

ATP EC_50_ values of current responses have now been added for the wt P2X7 receptor and key mutants F11*, S124*, K127*, D423*, A564* as well as the F11*/S124C double mutant. EC_50_ values at the mutants differed less than 3-fold from that at the wt. These data have been included in Table 2, Figure 3—figure supplement 3, and Figure 5—figure supplement 2 and were referred to in lines 208, 271, 315.

When studying fluorescence changes at the level of the head domain (p7), the authors write that "fluorescence and current changes started simultaneously and showed shorter rising times upon repeated ATP applications"; "K127* showed a fluorescence change that was faster than the current increase" but no quantification of the fluorescence and current kinetics is given. It would be nice to have such quantification. It would be also nice to have a comparison of the fluorescence and current ATP dose-response curves for mutants showing sufficiently large fluorescence changes. This would further support the fact that ANAP at the level of the head domain tracks gating.

ATP dose response curves for currents have been added (see above). Generation of dose response curves for fluorescence was hampered by bleaching and varying background fluorescence of individual oocytes (see also above and Essential Revision, point 3).

Quantification of current and fluorescence kinetics for the key mutants F11*, S124*, K127* and the TMRM-labeled F11*/S124C double mutant has now been performed and the respective data have been included in Figure 3—figure supplement 3 (see also Essential Revision, point 2).

ANAP incorporation at TM2 positions seems to induce constitutive currents in P2X7Rs. Is there a channel blocker specific for P2X7Rs that would allow to give an estimation of constitutive activity of these mutants?

To our knowledge, there is no channel blocker of P2X7 available. However, the P2X7 antagonist A438079 (Allsopp et al., 2018) was able to decrease the leak currents of the T340* and L341* mutants. These data are now included as Figure 4—figure supplement 1 (referred to in line 223).

Page 9, it is written that "the kinetics of F11* fluorescence correlated with current facilitation". As before, this statement solely relies on the qualitative observation of current and fluorescent traces. Quantitative measurements of fluorescence and current kinetics during the first and second application of ATP is necessary to substantiate that claim. Here again, a dose-response curve of current and fluorescence for ATP of P2X7-F11* would be a plus to determine the gating step(s) tracked by ANAP at this position.

These data have now been included (see above and Essential Revision, point 2).

Attributing the lack of ATP-induced fluorescence changes observed in mutants with ANAP incorporated in the ballast domain to a lack of functional coupling between the extracellular and the intracellular domains is a strong assumption. The issue with ANAP incorporation in the C-terminus is the membrane expression of truncated proteins that are functional. Since the authors showed that these truncated proteins exists, if they are functional, they contribute to the measured current. The proportion of ANAP-labeled proteins could thus be too low to yield fluorescence changes despite large currents. To study the contribution of truncated proteins, the authors should characterize these truncated proteins by expressing just the mRNA with the amber stop codon in *Xenopus* oocytes, investigate potential ATP-induced currents carried by these truncated receptors and measure ATP dose-response curves. Comparing ATP dose-response curves of ANAP-mutants to purely truncated proteins should allow estimating the contribution of truncated proteins to the general current.

Thank you for the helpful suggestion, these control experiments have now been performed for the D423* or A564* mutants and revealed less than 5% of the respective control current responses (RNA co-injected with all essential components) if the respective RNAs were co-injected only with water.

Thus, these truncated subunits do not or very inefficiently form functional homotrimers and their contribution to the current can be largely excluded. However, they might form functional heterotrimers with ANAP-labeled full-length subunits and as such, could still contribute to the fluorescence signal. Nevertheless, the fact that the EC_50_ values at the ANAP-labeled constructs were similar to those at the wt P2X7, argues against a major effect of the truncated subunits.

These data have now been included as Figure 5—figure supplement 2.

It seems that calcium induces fluctuations of ANAP fluorescence. Does calcium have any direct effect on ANAP fluorescence?

To test this possibility, we measured the emission spectra of ANAP-containing recording solution in the absence and presence of 10 μM, 100 μM, 1 mM, 10 mM, 100 mM, or 1 M Ca^2+^ and found that Ca^2+^ increases ANAP fluorescence, but only at concentrations above 100 mM. We included these photometric data in Figure 6—figure supplement 1.

It needs to be considered, however, that the Ca^2+^-induced fluctuations in VCF-measurements occurred also in the absence of ANAP (Figure 6—figure supplement 1).

It seems that fluorescence changes at some positions of the extracellular head domain and intracellular N-terminus track the process of facilitation. Tracking of this process is probably the most interesting finding of this paper, since the mechanism of facilitation is not known. In the paper, facilitation is measured as the increase of channel opening kinetics upon multiple applications of ATP. However, the first application of ATP clearly shows a small, fast component and a bigger, slow component (see, for instance, Figure 3F, 5C, 6A). Could it be that the first component reflects initial gating and the second one the slow facilitation process that occurs upon prolonged ATP application? Could the fast kinetics of K217* fluorescence match the fast component of current, indicating that K217* tracks the initial ATP opening step but not current facilitation? Similarly, on Figure 6 it looks like TMRM kinetics match first the fast current component, while F11* fluorescence matches the kinetics of current facilitation. Could the authors comment on it?

Indeed, current changes appear to be biphasic and we cannot exclude that the fast current component from K127* and TMRM-labeled F11*/S124C P2X7 match initial fast gating that occurs before the onset of facilitation and is reported by ANAP in position 127 or TMRM in position 124. However, the initial fast current component increases substantially between first and second ATP-application. This is not mirrored by the fluorescence signals, which are virtually identical between first and second ATP-applications. Therefore, we think that the fluorescence for this mutant reports a different process. Also, taking into account the proximity of the fluorophores to the orthosteric binding-site we consider it more likely, that these mutants report a process related to ligand binding. Compare also new Figure 3—figure supplement 3.

Reviewer #2 (Recommendations for the authors):To strengthen the points of weakness above and to elevate the scientific merits of the paper, I have some comments which require attention.(1A) I am curious to know the P2X7 structural differences by VCF between just before the 1st application and just before the 2nd application of ATP, to clarify the structural background of "memory" of ATP application.

Comparing the structures before first and second ATP application is very difficult with an indirect method like VCF, which provides mainly information about dynamic processes. Determination of different receptor states with this method would require clearly distinct fluorescence signals upon first and second activation, detailed structural information, and most likely numerous additional mutants to deduce and confirm specific conformational changes. Based on our observations, we generally suggest a potential transition into a pre-activated state (line 446) but do not want to elaborate on this to avoid overinterpretation.

(1B) Will there be any difference in (1A), in the presence and absence of extracellular Ca^2+^, i.e. Ca^2+^ influx via P2X7.

Experiments in presence of extracellular Ca^2+^ (1 mM) were performed to test for Ca^2+^-specific effects (Figure 6—figure supplement 1) but revealed large unspecific signals (even in the absence of ANAP) in the spectral range of ANAP emission and thus hampered analysis. Recordings in 0.5 mM Ca^2+^ from oocytes expressing F11* and S124* did not reveal obvious changes in the pattern of ANAP fluorescence signals compared to measurements in divalent-free recording solution.

(2A) Facilitation of F change was detected e.g. in S124*, F11*, S127C. Possible mechanisms of facilitation are discussed including mediators such as GDP, cholesterol, palmitoylation, Ca^2+^/CaM (Lines 335-342), However, there are no experimental results to support or deny them. It is desired to examine the effect of experimental manipulations of these factors on the F(and I) facilitation (e.g. removal of extracellular Ca^2+^).

We performed additional experiments to investigate effects of palmitoylation (using two mutants in which the palmitoylation sites were removed, Figure 3—figure supplement 3) and a CaM interaction using a mutant with mutated CaM-binding motif and investigating FRET between ANAP-labeled P2X7 and mNeonGreen-tagged CaM (Figure 5—figure supplement 3 and Figure 6—figure supplement 3, respectively). These data confirmed that deletion of a cysteine-rich intracellular region eliminates current facilitation (Roger et al., 2010) and that some of our mutants indeed track facilitation. However, mutation of the CaM binding site and FRET experiments did not support an effect of calmodulin or were inconclusive. Please see Essential Revisions points 1and 2 for details.

We would also like to point out that the changes observed in our study are compatible with a facilitation process that is an intrinsic property of the channel and does not require intracellular factors.

(2B) Line 273: "the so-called current facilitation in P2X7 is due to a change in receptor gating rather than ligand binding", Line 366: "Ligand binding is unaltered between the 1st and 2nd activation and consequently, changes in channel gating account for the observed current facilitation". I wonder if this is correct, as F facilitation is observed in S124* which is close to the ATP binding site (Figure 3F).

As described, the fluorescence changes can report both ligand interactions and protein movements as well as combinations of both. The specific effects depend not only on the orientation and position of the mutated residue but also the fluorophore used (e.g. different kinetics of TMRM and ANAP in position 124, compare also line 420ff). The fact that some mutants, like TMRM-labeled double mutant F11*/S124C and K127* detect fast processes (likely associated with direct ligand interactions or movements during ligand binding), does not exclude the possibility that fluorophores in neighboring positions (like ANAP-labeled S124*) detect relative domain movements within the receptor (likely resulting from a quenching/dequenching by aromatic residues or changes in the polarity of the environment) that contribute to the opening mechanism.

(2C) F facilitation was detected not in S124C-TMRM (Figure 6A), not in K127* (Figure 3F), but in S124* (Figure 3F). I cannot fully understand the ground of differences.

ANAP can detect localized changes in its environment in a distance of about 5 Å. Therefore, the immediate environment can be completely different for S124* and K127* and ANAP can even report different changes in neighbouring positions. In case of ANAP-labelled S124* and TMRM-labeled S124C, the very differently sized fluorophores most likely sense different environments and therefore also report different interactions/effects (quenching or polarity).

(3) As to the function of the ballast domain, it is described "intracellular ligand or interactors are required to "activate" this domain" (line 421). A possible involvement of the ballast domain to the functional modulation of P2X7 by the binding of various factors such as GTP/GDP, LPS and Ca^2+^/CaM are discussed (lines 407-415). The authors showed limited structural rearrangements of the ballast domain upon ATP application, but there is no evidence by VCF to prove activation by intracellular ligand. I would like to see data of F change of such as A564* in the ballast domain in response to manipulation of intracellular factors.

In contrast to patch-clamp experiments, direct manipulation of intracellular factors during the recording is difficult if not impossible in TEVC experiments. As an alternative, we generated mutations (as suggested) in the palmitoylation site and the CaM binding site and performed FRET experiments with co-expressed ANAP-labeled P2X7 and mNeonGreen-tagged CaM. We further did experiments in different concentrations of extracellular Ca^2+^. For details, please see Essential revision points 1 and 2. These data are now included in Figure 3—figure supplement 3, Figure, Figure 5—figure supplement 3, Figure 6—figure supplement 1, and Figure 6—figure supplement 3.

(4) Figure 4C: Does the F facilitation in T340* truly reflect "facilitation", in spite of the absence of current facilitation (as described in line 376)? The authors explained that a lack of faster current in the 2nd application is due to "strongly facilitated gating" (line 377). If that is the case, why there is F facilitation?

The beauty of VCF is the ability to differentiate between electrophysiologically defined receptor states (open and closed) and associated domain movements. It is well possible, that due to a "leaky" gate "naive" and facilitated receptor states cannot be resolved electrophysiologically, while the associated domain movements can still be detected. In principle, the mechanism of the hinge (gate movement measured by fluorescence) does not change when the door (gate, measured electrophysiologically) has a hole.

5) Do the F change of R-Geco (Figure 6B) and FRET change by mNeonGreen (Figure 6C) disappear in the absence of extracellular Ca^2+^?

R-GECO is a widely used Ca^2+^-sensor and fluorescence changes were not expected in the absence of extracellular Ca^2+^. Since the FRET approach was discontinued, these control experiments were not performed.

Reviewer #3 (Recommendations for the authors):1. Please cite this paper when talking about eRF-E55D as well, https://doi.org/10.7554/eLife.37248.

Thank you, done (line 103).

2. The 3'-terminal CCA-sequence is already in the pANAP plasmid. More elaboration on how it augmented the translation is needed to make it clear to readers.

The 3’-terminal CCA sequence is important for aminoacylation and translation. The pANAP sequence described in (Chatterjee and Guo, 2013) in the Supporting Information S2 encoding the tRNA lacks a 3’-CCA. This sequence might be dispensable, as it would likely be added by *X. laevis* oocyte-endogenous tRNA-nucleotidyltransferases. However, we decided to add it, to provide a functional tRNA ready for aminoacylation by the ANAP-specific tRNA synthetase already upon cytoplasmic injection. To avoid misunderstanding, we replaced "augmented" by "provided".

3. Additional FRET experiments might be out of the scope of this paper, but please add more discussion and references regarding this possibility in the paper.

We included several references to illustrate the potential of ANAP as a FRET partner more clearly (lines 328ff).

4. One area that could be improved on is the uncoupled "ballast" domain. How might GTP/GDP and dinuclear Zn2+-binding rearrange this domain? Can pharmacological manipulations be used to change intracellular GTP/GDP or Zn2+ levels to see how Anap labeled in this domain changes its fluorescence differently? Could the palmitoylation of the anchor domain play a role for uncoupling the "ballast' domain and for the current facilitation?

We are not aware of any protocols to change intracellular GTP or Zn^2+^ levels in a time scale that is amenable to our VCF analysis. As described above, we analyzed the Cys-free mutants that are likely not palmitoylated. These analyses confirmed the importance of palmitoylation for the facilitation process and are now included (see Essential revision point 1). However, expression of these mutants was too low for VCF analysis and no obvious effects of palmitoylation on fluorescence amplitude or kinetics could be detected. This is now mentioned in lines 190-205.

5. Line 634, should be "artifacts"?

Thank you, corrected!